# Adversarial Data Robustness via Implicit Neural Representation

## Abstract

Despite its effectiveness, adversarial training requires that users possess a detailed understanding of training settings. However, many common users lack such expertise, making adversarial training impossible and exposing them to potential threats. We propose "adversarial data robustness", allowing the data to resist adversarial perturbations. Then, even if adversaries attack those data, these post-attack data can still ensure downstream models' robustness at users' end. This leads to our new setup, where we store the data as a learnable representation via Implicit Neural Representation (INR). Then, we can train such a representation adversarially to achieve data robustness. This paper analyzes the possible attacks to this setup and proposes a defense strategy. We achieve a comparable robustness level without resorting to model-level adversarial training.

## 1 Introduction

Adversarial training, a common defense mechanism against adversarial attacks, operates at the model level by specifically training deep learning models with adversarial examples. However, model-level adversarial training may undermine the desired performance of deep learning models when the robustness requirement is incorporated. Moreover, introducing large-scale base models also makes model-level adversarial training difficult. Therefore, it is essential to investigate alternatives to enhance these models' robustness without resorting to model-level adversarial training.

As adversarial perturbations are directly injected to images (Szegedy et al., 2014; Madry et al., 2018), a straightforward solution is to make those images robust to adversarial attacks, rather than optimizing a model adversarially like existing solutions (Madry et al., 2018; Zhang et al., 2020). This way, even when adversaries corrupt images as usual, ready-to-use models accessible to ordinary users can continue functioning properly. We call this **adversarial data robustness**, and propose "training data adversarially" to achieve this goal. For images, one straightforward approach is to consider image pixels as learnable parameters and optimize these learnable parameters adversarially. However, such a straightforward solution significantly undermines image quality since the pixel values are directly manipulated. Therefore, we employ the Implicit Neural Representation (INR) as the cornerstone for our purposes. INRs represent data by optimizing a neural network to continuously map the coordinates to the corresponding data values (Dupont et al., 2021; Mildenhall et al., 2020; Sitzmann et al., 2020; Chen et al., 2022). Once the optimization converges, image pixels are stored as network weights, and the user can query neural networks with the corresponding coordinates to recover the whole image. Moreover, such a network-based representation framework is compatible with classical adversarial optimization for networks, enabling the generalization of existing model-level adversarial defense techniques.

We envision a **setup** for adversarial data robustness where typical model users or even the model developers are not burdened with extensive modifications to established frameworks. In this setup, images are pre-converted to their respective INRs and then sent to the model users. These users can then invoke straightforward functions encapsulated within an API to obtain the images for subsequent tasks. Since the generation process of representations incorporates adversarial robustness, images obtained from the post-attack representations maintain stable performance for downstream applications. For example, when the modifications to deep learning models are not permitted, or such modifications are costly, the model developers are not capable of training the model adversarially. By adversarially training the data, the adversarial data robustness can guarantee the function-

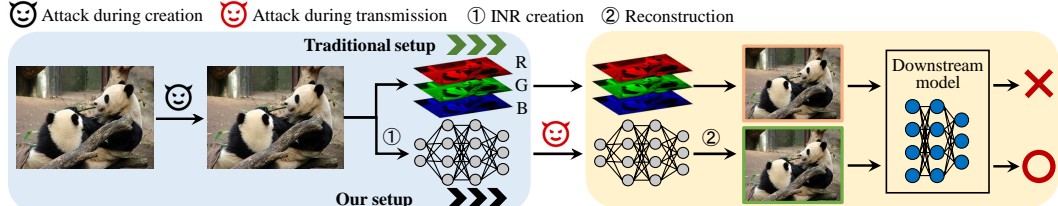

Figure 1: In the traditional setup, images can be manipulated by adversarial perturbations to fool downstream models. In our setup, we represent images as INRs. By applying a defense strategy at the INR creation stage, the data can defend against adversarial perturbations during data creation and transmission. Downstream models' robustness to adversarial attacks can be ensured without adversarial training for models.

ality of the downstream model without altering the model's parameters. This situation has become common, especially when downstream tasks use frozen large models (Guo et al., 2023), as their weights are difficult to modify.

We first explore possible attack types to the INR used as the data representation. Since only INR representation is used during the data transmission, an attack approach is to add adversarial perturbations to the network parameters for data representation. We, therefore, consider two strategies: 1) **Attack during creation**, which transfers adversarial perturbations from post-attack images to network parameters; and 2) **Attack during transmission**, which directly injects perturbations into the representation during its transmission. Given that the network parameters of INR are more susceptible to alterations due to external perturbations (Shu & Zhu, 2019), our emphasis is on balancing between attack efficacy and the quality of images generated from post-attack representations. We propose Double Projection Gradient Descent (DPGD) as a method for conducting attacks during data transmission, which can achieve a better balance by implementing gradient constraints on the image during the backpropagation process.

We propose defense-in-creation to defense above attacks within this setup. As Figure 1 depicts, our goal is to achieve adversarial data robustness in the creation process of INR. The INR generated by our method can defend against attacks during creation and attacks during transmission, ensuring the performance of specific downstream applications. To generate robust INR, we solve a min-max optimization problem by incorporating a robustness loss on top of the existing INR training framework. The weights assigned to these two losses can be used to strike a balance between reconstruction quality and robustness. Our research pioneers data robustness exploration, thereby charting a new course towards AI safety. Our major contribution can be concluded as follows:

- Our adversarial data robustness can guarantee the robustness of deep learning models against adversarial attacks, without necessitating model-level adversarial training.
- We examine potential attack types to INR used for the data representation. A Double Projection Gradient Descent (DPGD) is proposed to ensure adversarial patterns' invisibility when directly injected into INR parameters used for data representation.
- We have formulated a defense-in-creation strategy to defense the possible attacks.

## 2 RELATED WORK

**Implicit neural representations.** Implicit Neural Representation (INR) is a technique that leverages neural networks to create a mapping between a coordinate and its corresponding signal value. This method offers a continuous and memory-efficient way to model various signals, including 1D audio (Gao et al., 2022), 2D images (Tancik et al., 2020), 3D shapes (Park et al., 2019), 4D light fields (Sitzmann et al., 2021), and 5D radiance fields (Mildenhall et al., 2020). Direct supervision can be used to train accurate INR models for these signals by comparing the network output to ground truth data or indirect means, such as calculating the loss between the output after differentiable operators and a variant of the ground truth signal. This makes INR a powerful tool for solving inverse problems by taking advantage of the well-known forward processes in these problems. INR has found widespread applications in various fields, including computer vision and graphics (Tewari et al., 2022), computational physics (Karniadakis et al., 2021), biomedical engineering (Liu et al.,

2022; Zhu et al., 2022), material science (Chen et al., 2020), and fluid mechanics (Raissi et al., 2020; Reyes et al., 2021). As INR is compatible with adversarial training framework, we propose to use INR as cornerstone to achieve data-level robustness.

**Adversarial attack and defense.**   Adversarial attacks and defense have emerged as significant research areas in machine learning. To explore the vulnerability of DNNs, various attacks have been proposed (Carlini & Wagner, 2017; Croce & Hein, 2020; Goodfellow et al., 2015; Madry et al., 2018), which can be generally categorized into white-box attacks and black-box attacks (Goodfellow et al., 2015). Most white-box attacks use gradients to obtain the perturbations on the inputs which maximize the loss function, including the Fast Gradient Sign Method (FGSM) (Szegedy et al., 2014), Projection Gradient Descent (PGD) (Madry et al., 2018) and Carlini and Wagner (CW) (Carlini & Wagner, 2017). Black-box attacks involve two methods, where attackers lack access to the victim models' information. Query-based methods approximate perturbations through a large number of queries (Andriushchenko et al., 2020; Cheng et al., 2018; Guo et al., 2019), while transfer-based methods utilize a surrogate model to generate adversarial examples with higher transferability (Liu et al., 2016; Wang & He, 2021). To counter these attacks, researchers have developed various defense mechanisms (Cohen et al., 2019; Li et al., 2021). One common approach is adversarial training, which has shown relative resistance to most existing attacks. The vanilla adversarial training strategy involves incorporating adversarial examples into the training data to create a min-max game during optimization (Madry et al., 2018). Numerous variants of adversarial training algorithms have been developed to improve the performance of adversarial robustness, including early-stopping strategy (Rice et al., 2020), TRADES (Zhang et al., 2019), FAT (Zhang et al., 2020), and CFA (Wei et al., 2023). Other techniques for adversarial defense include defensive distillation (Papernot et al., 2016), gradient regularization (Ross & Doshi-Velez, 2018), gradient masking (Folz et al., 2020), and input denoising (Liao et al., 2018). LINAC (Rusu et al., 2022) proposes to transforms images into INR to train downstream networks to improve the robustness. However, these defense methods, which improve robustness via adversarially training downstream networks, are not feasible for networks with unmodifiable parameters and could undermine the desired performance. Therefore, we propose a method to achieve adversarial data robustness to solve this problem.

## 3    ADVERSARIAL DATA ROBUSTNESS

### 3.1    OUR SETUP

Our setup is rooted in the reality that many model users often fail to understand the settings associated with adversarial training. Moreover, model-level adversarial training can potentially compromise the intended performance of deep learning models. Therefore, when the adversarial training to models are difficult for common users, we aim to make the data robust to adversarial perturbations. Then, those robust data can ensure robust deep learning models.

Given that adversarial training has been an effective defense mechanism against adversarial attacks, the robust data can only be created via adversarial training. We thus propose storing the image as a network-based representation and enhancing this representation's robustness to potential adversarial perturbations. As depicted in Figure 1, even if this representation is attacked during its creation or transmission, users can still restore its clean state, ensuring robust performance of downstream tasks. We consider Implicit Neural Representation (INR) as the cornerstone of our formulation. For an image $I$, its INR is given by a MLP $f_\theta : \mathbb{R}^2 \to \mathbb{R}^3$ that maps the input spatial coordinate $\mathbf{p} = (p_x, p_y) \in \mathbb{R}^2$ to its corresponding pixel value $I(\mathbf{p}) \in \mathbb{R}^3$ as $I(\mathbf{p}) = f_\theta(\mathbf{p})$. $\theta$ represents the MLP's trainable parameters, which leave access for our adversarial data robustness.

Generally, $f_\theta$ can only reach its optimum status via iterative optimization. The optimization can be finished by overfitting $f_\theta$ to the image by minimizing a reconstruction loss between true pixel values $I(\mathbf{p}_{m,n})$ and predicted values $f_\theta(\mathbf{p}_{m,n})$ as

$$\mathcal{L}_{recon} = \frac{1}{M \times N} \sum_{m,n} \|f_\theta(\mathbf{p}_{m,n}) - I(\mathbf{p}_{m,n})\|_2^2, \tag{1}$$

where $(m, n)$ is the index of the corresponding pixel location, and $(M, N)$ is size of the image. After optimization, the image can be stored or transmitted as network weights $\theta$.

Figure 2: Samples after attack during creation. (a) are the ground truth images. (b) are the results of (a) being attacked by PGD. (c) are the images reconstructed from INRs optimized for (b). (d) are the residuals between (b) and (c). The output results of the classifier are labeled in the top left corner of each image. The images reconstructed from the representations attacked during their creation are almost the same to the images directly attacked by PGD, with a low average MSE value.

Once such a representation is received by model users, they can obtain the image $\hat{I}$ from this representation by querying INR $f_\theta$ at each pixel coordinate as $\hat{I}(\mathbf{p}_{m,n}) = f_\theta(\mathbf{p}_{m,n})$, where $(m, n)$ is the pixel index on the image. We define $\Psi : f_\theta \mapsto \hat{I}$ as the mapping from INR to the reconstructed image. The image $\hat{I}$ is then fed into the downstream model to obtain the result.

In light of this setup, we examine the potential attack types to INR. Subsequently, we will delve into the associated adversarial training to ensure data robustness.

## 3.2 ATTACK TYPES TO INR

An adversarial attack is to perturb the data used for downstream tasks (Szegedy et al., 2014). Usually, it is performed by introducing adversarial perturbations to input data, which is a designed slight alteration to data that can significantly change the output of downstream models. Since the data has been implicitly represented in our setup, the only way to conduct an adversarial attack is to inject adversarial perturbations into the representation. Similar to an adversarial attack directly for images, two criteria should be guaranteed: 1) **invisibility**: the perturbations are subtle and remain invisible within the content (Laidlaw et al., 2021); and 2) **attacking efficacy**: the invisible perturbations can still impair the efficacy of downstream applications (Szegedy et al., 2014). On the basis of these criteria, we explore attack types that can introduce adversarial perturbations to the data representation.

**Attack during creation.** The first option is to attack the representation by transferring adversarial perturbations from corrupted images to the respective network parameters of INRs. In this case, adversaries introduce perturbations to images and subsequently strive to impart these perturbations onto the network parameters of the INR during its construction. Consequently, the adversarial perturbations, when recovered from the representation along with the image content, intensify the loss function for downstream tasks (*e.g.*, cross-entropy loss for image classification).

Formally, the adversarial perturbation for the $i$-th image $I_i$ can be modeled as the solution of the following optimization problem, which is the same as the classical adversarial attacks (Madry et al., 2018):

$$\max_{\Delta I_i} \quad \mathcal{L}_{CE}\left(g_\phi\left(I_i + \Delta I_i\right), y_i\right), \quad \text{s.t.} \ \|\Delta I_i\|_p \leq \epsilon, \tag{2}$$

where $g_\phi$ denotes the neural network for downstream tasks (*e.g.*, classification (Szegedy et al., 2014)), $\Delta I_i$ is the adversarial perturbation for the $i$-th image, $y_i$ is the true label of $I_i$ for classification task, and $\mathcal{L}_{CE}(\cdot)$ denotes the cross entropy loss function. The $\|\cdot\|_p$ denotes the $\ell_p$-norm, and the constraint limits the size of perturbation with maximum allowable value $\epsilon$ to avoid being recognized by humans. Unless otherwise stated, the infinite norm is adopted by setting $p = \infty$ in this paper following previous works (Dong et al., 2023).

INRs can accurately capture the very fine details of images (Chen et al., 2021; Strümpler et al., 2022). Therefore, during the creation of an INR, the INR weights can accurately represent adversarial perturbation $\Delta x$. The reconstructed image from such an INR retains the adversarial information and can deceive the downstream applications. As shown in Figure 2, the images reconstructed from INR under attack during creation resemble the corrupted images with adversarial perturbation, allowing the reconstructed image to fool the downstream applications.

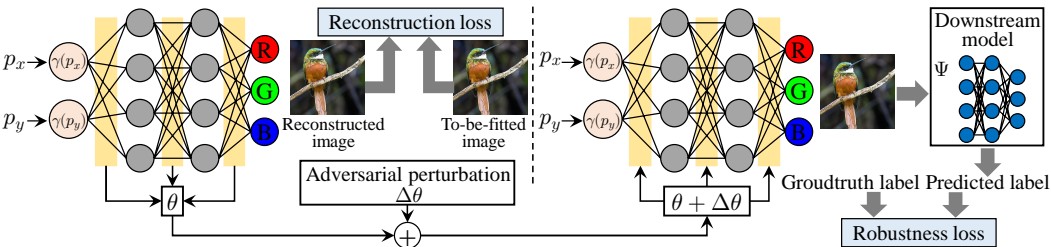

Figure 3: Framework of defense-in-creation. We encode an image into the parameter of its corresponding INR, $\theta$. **Left**: we calculate the distance between the reconstructed image and the to-be-fitted image to establish a reconstruction loss for high-quality reconstruction. **Right**: we introduce adversarial perturbation into the parameter $\theta$ during INR creation process, and design a robustness loss to ensure downstream models' robustness by comparing its prediction with the groundtruth.

**Attack during transmission.** The second strategy is to attack the representation directly. This attack typically occurs during the transmission of the data representation. When the adversaries access the representation, they can directly inject adversarial perturbations into the network parameters. The process of conducting an adversarial attack to the INR representation of $i$-th image during its transmission can be concluded as follows:

$$\max_{\Delta\theta_i} \quad \mathcal{L}_{CE}\left(g_\phi\left(\Psi\left(f_{\theta_i+\Delta\theta_i}\right)\right), y_i\right), \quad \text{s.t. } \|\Delta\theta_i\|_p \leq \delta, \tag{3}$$

where $\Delta\theta_i$ is the adversarial perturbation for the $i$-th image INR, and we also use $\ell_\infty$-norm in Equation 3 to limit the size of the perturbation.

A simple method to conduct attack during transmission is to manipulate the INR parameters using the traditional gradient-based attack methods, such as Fast Gradient Sign Method (FGSM) (Szegedy et al., 2014), Projected Gradient Descent (PGD) (Madry et al., 2018), and Carlini and Wagner (CW) (Carlini & Wagner, 2017). These attacks are referred to as FGSM for INR, PGD for INR, and CW for INR, respectively. While these methods are initially designed to operate on images, integrating INR into the differentiable pipeline allows us to derive the gradient value of INR parameters by leveraging the loss value of the downstream classifier. The whole process of PGD attack for INR is shown in Algorithm 1. Given an optimized INR, we start with randomly perturbed parameters within the maximum perturbation. In every iteration, we reconstruct the image from the INR and feed it into the classifier to compute the loss. Then, we iteratively apply gradient updates and ensure that the perturbed parameters remain within the allowable range through projection. However, as INR parameters are sensitive to direct manipulations, such a straightforward solution can easily undermine the quality of images obtained from the post-attack representation.

To better balance the attacking efficacy and the image quality, we propose a **Double Projection Gradient Descent (DPGD)** that aims to directly manipulate the INR parameters while preserving the reconstructed image's quality. The detailed process of DPGD can be found in Algorithm 2. Unlike PGD for INR, we incorporate an additional projection in each iteration. This projection involves projecting the gradient backpropagated to the reconstructed image onto a boundary controlled by the factor $\zeta$. The motivation of this step is to directly limit the difference between the reconstructed image and the original image at the pixel level. For detailed derivation, please refer to Section B in appendix. The projected gradient is then further backpropagated to update the INR parameters. The quality of images derived from the representations remains high by constraining the gradient on the reconstructed image. The factor $\zeta$ can control the balance between the attacking efficacy and the image quality.

### 3.3 DEFENSE-IN-CREATION

To achieve data-level robustness, we propose a defense-in-creation strategy that generates robust INR during the creation process to defend against adversarial attacks effectively. The perturbations generated in attacks during creation can be eliminated in optimizing phase, and the INR derived using defense-in-creation can further defend against potential attacks during transmission. By employing this approach, we can ensure the performance of downstream networks at a data level. Besides, since the basic function of INR is to store the content of images, we also consider the quality of the reconstructed image in our method.

---

**Algorithm 1** PGD for INR

---

**Input**: Optimized INR $f_\theta$, ground truth label $y$, and pre-trained downstream model $g_\phi$
**Parameter**: Number of iterations $N$, step size $\alpha$, maximum perturbation $\delta$
**Output**: Parameters of adversarial INR $\theta_{adv}$

1: Let $\theta_{adv} = \theta$.
2: **Random start**: $\theta_0 = \theta_{adv} + \Delta\theta_0$, $\theta_0 \sim \mathcal{U}(-\delta, \delta)$.
3: **for** $t = 1, 2, \cdots, N$ **do**
4:    Reconstruct image $\hat{I}_{t-1} = \Psi(f_{\theta_{t-1}})$;
5:    $\theta_t = \theta_{t-1} + \alpha \cdot \text{sign}(\nabla_{\theta_{t-1}}\mathcal{L}_{CE}(g_\phi\left(\hat{I}_{t-1}\right), y))$;
6:    Update $\theta_t = \text{Proj}_{\|\theta_t - \theta\|_p \leq \delta}(\theta_t)$.
7: **end for**
8: **return** $\theta_{adv} = \theta_N$

---

**Algorithm 2** Double Projection Gradient Descent (DPGD)

---

**Input**: Optimized INR $f_\theta$, ground truth label $y$, and pre-trained downstream model $g_\phi$
**Parameter**: Number of iterations $N$, step size $\alpha$, maximum perturbation $\delta$, gradient control factor $\zeta$
**Output**: Parameters of adversarial INR $\theta_{adv}$

1: Reconstruct original image $\hat{I}_{org} = \Psi(f_\theta)$;
2: Let $\theta_{adv} = \theta$.
3: **Random start**: $\theta_0 = \theta_{adv} + \Delta\theta_0$, $\theta_0 \sim \mathcal{U}(-\delta, \delta)$.
4: **for** $t = 1, 2, \cdots, N$ **do**
5:    Reconstruct image $\hat{I}_{t-1} = \Psi(f_{\theta_{t-1}})$;
6:    $Grad_I = \frac{\partial \mathcal{L}_{CE}(g_\phi(\hat{I}_{t-1}), y)}{\partial \hat{I}_{t-1}}$;
7:    Update $Grad_I = \text{Proj}_{\|\hat{I}_{t-1} + Grad_I - \hat{I}_{org}\|_p \leq \zeta}(Grad_I)$; ▷ Projecting gradient of image pixels
8:    $Grad_\theta = \frac{\partial \hat{I}_{t-1}}{\partial \theta_{t-1}} \cdot Grad_I$;
9:    $\theta_t = \theta_{t-1} + \alpha \cdot \text{sign}(Grad_\theta)$;
10:   Update $\theta_t = \text{Proj}_{\|\theta_t - \theta\|_p \leq \delta}(\theta_t)$.      ▷ Projecting gradient of INR parameters
11: **end for**
12: **return** $\theta_{adv} = \theta_N$

---

As shown in Figure 3, defense-in-creation is implemented by designing a robust loss on top of the original reconstruction loss defined in Equation 1. The robustness loss is obtained by adding adversarial perturbations to the INR parameters and calculating the loss after feeding the reconstructed image into the classifier $g_\phi$. The whole defense process can be formulated as follows:

$$\min_\theta \quad \lambda_1 \mathcal{L}_{recon} + \lambda_2 \max_{\Delta\theta} \mathcal{L}_{CE}\left(g_\phi\left(\Psi(f_{\theta+\Delta\theta})\right), y_i\right), \quad \text{s.t. } \|\Delta\theta\|_p \leq \delta, \quad (4)$$

where $\Delta\theta$ is the adversarial perturbation added in the training phase. The parameters $\lambda_1$ and $\lambda_2$ balance the reconstruction quality and robustness. The detailed solution to the above optimization problem is shown in Algorithm 3. We use the robustness loss generated from Algorithm 1 to enhance the robustness, and employ reconstruction loss to guarantee the image quality. By minimizing the loss obtained from classifier $g_\phi$, we can effectively remove the adversarial perturbation embedded in the target image when performing attack during creation. By introducing adversarial samples during the training process of INR, INR can gradually learn to have better robustness against parameter perturbations, thereby defending against attack during transmission.

## 4 EXPERIMENTS AND RESULTS

### 4.1 EXPERIMENTAL SETTINGS

**Dataset.** We follow established settings (Zhang et al., 2020; Jin et al., 2023) on adversarial attack and defense to evaluate our proposed mechanism. Our experiments are conducted on three real-world datasets: CIFAR-10 (Krizhevsky et al., 2009), CIFAR-100 (Krizhevsky et al., 2009), and SVHN (Netzer et al., 2011). CIFAR-10/100 dataset contains $60K$ color images with the size

---

**Algorithm 3** Proposed defense method to produce INR against adversarial attacks

---

**Input**: Image $I$, ground truth label $y$, and pre-trained downstream model $g_\phi$
**Parameter**: Number of iterations $T$, learning rate $\eta$
**Output**: INR $f_\theta$ for image $I$

1: Initialize model parameters $\theta$.
2: **for** $t = 1$ to $T$ **do**
3:    Generate INR adversarial example $f_{\theta_{adv}}$ using Algorithm 1;
4:    Compute reconstruction loss $\mathcal{L}_{recon}$ as Equation 1;
5:    Compute robustness loss as $\mathcal{L}_{CE}\left(g_\phi\left(\Psi(f_{\theta_{adv}})\right), y\right)$;
6:    Compute the total loss $\mathcal{L}_{total}$ as Equation 4;
7:    Update parameters using adversarial example: $\theta \leftarrow \theta - \eta \cdot \nabla_\theta \mathcal{L}_{total}$
8: **end for**
9: **return** Trained INR parameters $\theta$

---

of $32 \times 32$, including $50K$ training images and $10K$ test images in 10 and 100 classes, respectively. SVHN is a dataset collected by Google Street View, consisting of $32 \times 32$ color images of house numbers, with $73,257$ training images and $26,032$ test images in 10 classes. The classifier is trained using training images. We train INRs for test images to evaluate our attack approaches and defense methods. As suggested in previous work (Wei et al., 2023; Jin et al., 2022), we evaluate the performance of attacks and defenses based on the PreActResNet-18 (He et al., 2016) and WideResNet34-10 (Zagoruyko & Komodakis, 2016) architectures on CIFAR-10/100 and SVHN.

**Implementation Details.** We implement our method using PyTorch. The INR is a 5-hidden layer MLP with 256 channels per layer and ReLU activation functions for all the data. Following Mildenhall et al. (2020), we also use a positional encoding for pixel coordinates with 5 frequencies. For adversarial defense training, 10-step PGD attack for INR is applied, with maximum perturbation size for parameters $\delta = 0.0006$. We use the Adam optimizer with default values and a learning rate of 0.001. A cosine learning rate decay schedule is used with the minimum value of the multiplier 0.0001. We optimize each INR for 1000 iterations. Other hyper-parameters are specified in each experiment. We conduct experiments on FGSM (Szegedy et al., 2014), PGD (Madry et al., 2018), Carlini and Wagner (CW) (Carlini & Wagner, 2017), and AutoAttack (AA) (Croce & Hein, 2020) for attack during creation. The maximum perturbation size $\epsilon$ is set to $8/255$ for FGSM and PGD. PGD is used with steps 100, and step size $0.2/255$. We apply FGSM for INR, PGD for INR, CW for INR, and DPGD for attack during transmission. The steps of PGD for INR, and DPGD are set to 100, and maximum perturbation size for INR parameters $\delta = 0.0006$, with step size $2.5 \cdot \delta/\text{steps}$ in all experiments. CW attack is applied by optimizing 100-step PGD (Zhou et al., 2022; Huang et al., 2021). All the experiments are performed on NVIDIA Tesla V100 GPUs.

**Baselines.** We found no method specifically for enhancing the robustness at the data level. Therefore, we compare with other three settings for comparison: 1) **Normal training**: optimizing the INR with only reconstruction loss; 2) **Direct pixel manipulation** [1]: direct manipulating the pixel values without INR encoding; 3) **Model-level adversarial training**: adversarial training for model robustness including AT (Madry et al., 2018), TRADES (Zhang et al., 2019), FAT (Zhang et al., 2020), GAIRAT (Zhang et al., 2021), and LAS-AT (Jia et al., 2022). Although the latter method aims to enhance the robustness through training the downstream classifier, we conduct a comparison with them, and the analysis can be found in Section 4.3.

**Evaluation metrics.** We assess all methods' performance by evaluating image quality, attack effectiveness, and defense ability. For image quality, we evaluate the performance by using PSNR, SSIM, and LPIPS (Zhang et al., 2018). Higher PSNR and SSIM value indicates better performance, while lower LPIPS value indicates better performance for both attack and defense. To assess the effectiveness of attacks, we utilize the attack success rate (ASR). In terms of defense, we evaluate the defense ability by considering the classification accuracy of the downstream classifier.

## 4.2 EXPERIMENTAL RESULTS FOR DIFFERENT ATTACKS

This section evaluates the performance of the attacks used in our research. Specifically, we assess the attacks during creation and transmission to evaluate their invisibility and attacking efficacy.

---

[1]Please refer to Section A.4 for more information.

Table 1: Evaluation of attack during creation with different image-based attack methods. The results are averaged on all examples in CIFAR-10 test dataset.

| Method | Clean images | Clean INR | Method | FGSM on images | Creation after FGSM | PGD on images | Creation after PGD | CW on images | Creation after CW | AA on images | Creation after AA |
|---|---|---|---|---|---|---|---|---|---|---|---|
| Accuracy | 94.97% | 94.73% | ASR | 60.45% | 59.26% | 100.00% | 99.83% | 99.93% | 99.84% | 100.00% | 99.89% |
| PSNR | - | 48.42 | PSNR | 30.14 | 31.01 | 32.70 | 33.70 | 32.68 | 34.12 | 32.71 | 33.85 |
| SSIM | 1.000 | 0.998 | SSIM | 0.921 | 0.935 | 0.956 | 0.967 | 0.957 | 0.970 | 0.956 | 0.968 |
| LPIPS | 0.000 | 0.002 | LPIPS | 0.146 | 0.134 | 0.082 | 0.071 | 0.082 | 0.067 | 0.082 | 0.070 |

Table 2: Reconstruction qualities and accuracies of downstream classifiers compared with normal training and direct pixel manipulation. The results are averaged on all examples in each dataset with different classifier architectures. Image quality evaluations are obtained by comparing with original clean images. The classifier architectures used for CIFAR-10/100 and SVHN are PreActResNet-18 and WideResNet34-10, respectively. More results can be found in the appendix.

| Dataset | Method | PSNR | Accuracy without attacks | Attack during creation | | | Attack during transmission | | | |
|---|---|---|---|---|---|---|---|---|---|---|
| | | | | PGD | CW | AA | DPGD | FGSM | PGD | CW |
| CIFAR-10 | Normal image | - | 94.97% | - | - | - | - | 39.55% | 0.00% | 0.07% |
| | Pixel manipulation | 33.16 | 100% | 100% | 100% | 100% | - | 99.88% | 33.80% | 27.88% |
| | Normal training | 48.38 | 94.73% | 0.17% | 0.16% | 0.11% | 20.80% | 57.36% | 18.84% | 17.72% |
| | **Defense-in-creation** | 42.92 | 100% | 100% | 100% | 100% | 99.60% | 100% | 98.68% | 61.36% |
| CIFAR-100 | Normal image | - | 76.92% | - | - | - | - | 8.44% | 0.01% | 0.00% |
| | Pixel manipulation | 29.63 | 100% | 100% | 100% | 100% | - | 99.96% | 30.08% | 0.56% |
| | Normal training | 48.26 | 76.44% | 0.16% | 0.12% | 0.11% | 4.32% | 17.68% | 2.16% | 1.76% |
| | **Defense-in-creation** | 40.50 | 100% | 100% | 100% | 100% | 98.96% | 100% | 98.44% | 74.52% |
| SVHN | Normal image | - | 95.80% | - | - | - | - | 41.84% | 1.32% | 0.96% |
| | Pixel manipulation | 35.78 | 100% | 100% | 100% | 100% | - | 100% | 76.60% | 18.24% |
| | Normal training | 53.03 | 95.80% | 1.28% | 1.20% | 0.18% | 39.04% | 65.96% | 36.24% | 35.76% |
| | **Defense-in-creation** | 46.98 | 100% | 100% | 100% | 100% | 99.88% | 100% | 99.88% | 84.84% |

**Attack during creation.** We validate the attack during creation, and the findings are presented in Table 1. We compare the results of this attack on the INR parameters with those obtained by directly inputting the images attacked by FGSM, PGD, CW, and AA into the downstream classifier. The reconstructed images display comparable attack effectiveness and image quality to those manipulated using FGSM, PGD, CW, and AA methods. Due to the excellent representation capabilities of INR, the performance of the attack during creation is determined by attacks applied to the images.

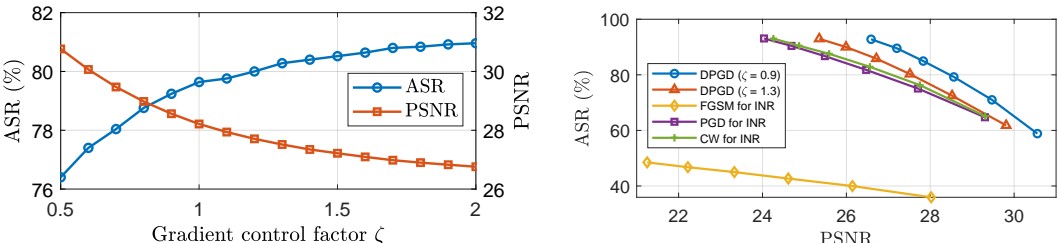

Figure 4: Evaluation of attack during transmission. Left: we present the ASRs and PSNRs across varying gradient control factors in DPGD. Right: we compare the ASR v.s. PSNR curves with varying the maximum perturbation across different methods. The results are averaged on all examples in CIFAR-10 test dataset.

**Attack during transmission.** We evaluate the impact of the image gradient control factor $\zeta$ on the invisibility and attacking efficacy in the DPGD algorithm. We fix the maximum perturbation at $\delta = 0.0006$, and the ASR and image quality results are shown in Figure 4(left). The results indicate that the factor $\zeta$ can regulate the balance between invisibility and attacking efficacy. Specifically, a larger value of $\zeta$ indicates higher attacking efficacy, while a smaller value of $\zeta$ indicates a closer resemblance to the original image. We further compare our DPGD algorithm with FGSM for INR, PGD for INR, and CW for INR in Figure 4(right). We take $\zeta = 0.9$ and $1.3$, respectively, and vary the maximum perturbation $\delta$ to obtain curves illustrating the relationship between image quality and ASR. At the same level of image quality, our method achieves higher ASR, indicating a higher attacking efficacy. Therefore, our proposed DPGD achieves better performance.

## 4.3 EXPERIMENTAL RESULTS FOR DEFENSE

In this section, we present experimental results of our proposed adversarial defense method and compare them with the baselines.

Table 3: Accuracies compared with model-level adversarial training. The classifier structures are all WideResNet34-10. The results are averaged on all examples in CIFAR-10 test dataset.

| Defense method | Defense-in-creation | AT | TRADES | FAT | GAIRAT | LAS-AT |
|---|---|---|---|---|---|---|
| Natural | 94.32% | 85.90% | 85.72% | 87.97% | 86.30% | 86.23% |
| PGD | 92.51% | 53.42% | 53.40% | 47.48% | 40.30% | 53.58% |

**Robustness and reconstruction quality.** We further evaluate our proposed adversarial defense for achieving data robustness. We set $\lambda_1 = 1$, $\lambda_2 = 0.0022, 0.0006, 0.001$ for CIFAR-10, CIFAR-100, and SVHN, respectively. The results are shown in Table 2. We study the performance of the proposed defense in defending against various attack methods. Our method achieves the highest accuracy and produces high-quality reconstructions across different attacks, datasets, and classifier architectures. While normal training achieves the best reconstruction results, it is vulnerability to all attacks, resulting in low accuracy under various attack scenarios. Although pixel manipulation can defend against attacks during creation, it leads to lower accuracy when encountering attacks during transmission, such as PGD for INR and CW for INR.

**Comparison with different $\lambda_2$.** To evaluate the balance between reconstruction quality and robustness in our defense-in-creation strategy, we adjust the weight ratio between reconstruction loss $\lambda_1$ and weight of robustness loss $\lambda_2$. We conduct a series of experiments under attacks during transmission by fixing $\lambda_1 = 1$ and varying the weight of robustness loss $\lambda_2$. As shown in Figure 5, a smaller value of $\lambda_2$ leads to higher reconstruction quality, while a larger value of $\lambda_2$ enhances robustness against attacks.

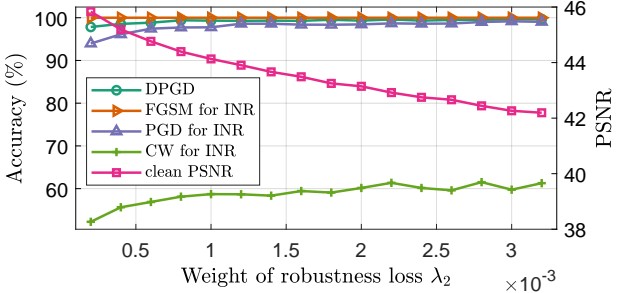

Figure 5: Balance between reconstruction quality and robustness against attacks during transmission. The results are averaged on all examples in CIFAR-10 test dataset.

**Comparison with model-level adversarial training.** Our method aims to enhance data-level adversarial robustness, and its motivation differs from traditional adversarial training approaches. As attacks on INRs are different from those on images, we apply PGD for INR in our method. Furthermore, to ensure a fair comparison, we use the predicted labels generated by the downstream classifier instead of the ground truth labels in the creation process of INR. As illustrated in Table 3, our method demonstrates better performance with natural data and stronger robustness compared with model-level adversarial training methods.

## 5 CONCLUSION

We demonstrate adversarial data robustness in this paper by utilizing implicit neural representations. Instead of adversarially training deep learning models, we focus on adversarially training the data. Doing so ensues that the data retains robust performance even if it is subjected to attacks before it reaches the model users. To accomplish this, we initially represent the data implicitly and focus on adversarial training of this representation during its formation. By analyzing the potential attacks, we unveil an adversarial training scheme that bolsters the robustness of implicit data representations against potential adversarial attacks. In our proposed approach, ready-to-use image classifiers exhibit adversarial robustness on par with models that undergo model-level adversarial training.

**Limitations.** Though our method is successful in achieving robustness at the data level, it cannot handle some extreme cases. For example, if malicious users gain access to the reconstructed images from the INR, they can directly apply adversarial perturbations to the image pixels, thereby circumventing attacks on the INR. This problem can be solved through system design by protecting the process from image reconstruction to downstream model input, to ensure that the recovered images are not accessible. Besides, we only consider risks caused by malicious technical designs. However, as security is not solely a technical problem, we still need to collaborate with various parties to consider the legal and social impact. We will further explore the feasibility of our approach in other data types and downstream tasks in the future.

ETHICS STATEMENT

Our goal is to ensure the security of deep neural networks. We did not employ crowdsourcing and did not involve human subjects in our experiments. When utilizing existing assets such as code, data, or models, we have properly cited the original creators.

REPRODUCIBILITY STATEMENT

We present the detailed network structure and some main hyper-parameter settings in Section 4.1. Other hyper-parameters are specified in each experiment. More experimental details can be found in Section A in appendix. We also provide a demo code in Supplementary Material. We plan to release the entire source code with random seeds for reproducibility at a later time.

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

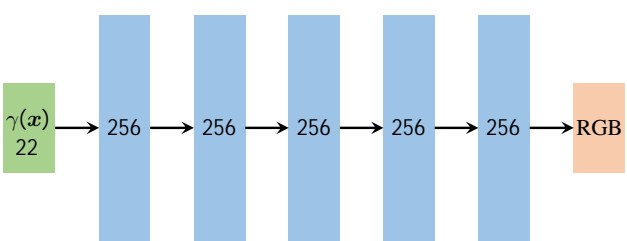

Figure 6: A visualization of our INR architecture.

# A    MORE IMPLEMENTATION DETAILS

## A.1    NETWORK STRUCTURE

We show the details our INR structure in Figure 6. Input vectors, hidden layers, and output vectors are depicted in green, blue, and orange, respectively. The number inside each block indicates the dimension of the corresponding vector. Following Mildenhall et al. (2020), we utilize a positional encoding technique to transform pixel coordinates into a higher-dimensional space, enabling us to effectively capture higher-frequency information. Every pixel coordinate, $d$, is first normalized to the range of $[-1, 1]$, and then subjected to the following transformation:

$$\gamma(d) = \left[\sin\left(2^0 \pi d\right), \cos\left(2^0 \pi d\right), \sin\left(2^1 \pi d\right), \cos\left(2^1 \pi d\right), \dots, \sin\left(2^{F-1} \pi d\right), \cos\left(2^{F-1} \pi d\right)\right]. \tag{5}$$

After concatenating the original coordinates with the results of positional encoding, the input is fed into the network. In all of our experiments, we employ $F = 5$ frequencies and utilize 5-hidden layer MLP, where each layer consists of 256 units and uses ReLU non-linearities.

## A.2    INR TRAINING DETAILS

The optimizer that we use for training the INR is Adam (Kingma & Ba, 2015), with default parameters and a learning rate of $\mu = 0.001$. A total of 1000 optimization steps are performed, with each step involving the entire set of pixels of the to-be-fitted image. We utilized a cosine learning rate decay schedule to enhance convergence, setting the minimum value of the multiplier $\alpha = 0.0001$ (Loshchilov & Hutter, 2017).

## A.3    DOWNSTREAM CLASSIFIERS

In our experiments, we should first train a classifier in a regular way. We select two typical classifier architectures, namely PreActResNet-18 (He et al., 2016) and WideResNet34-10 (Zagoruyko & Komodakis, 2016). We strictly follow the instructions in the original paper to construct the network. For both architectures, we utilize SDG as the optimizer, with an initial learning rate of 0.01, momentum of 0.9, and weight decay of $5e - 4$. The total training epoch is 200. The learning rate is multiplied by 0.2 at epoch $(60, 120, 160)$ respectively.

## A.4    DIRECT PIXEL MANIPULATION

Direct pixel manipulation is one of our baselines. In this method, we consider image pixels as learnable parameters and optimize these learnable parameters adversarially. The process of direct pixel manipulation can be formulated as a min-max game as follows:

$$\min_{I_m} \quad \lambda_1 \|I_m - I\|_2^2 + \lambda_2 \max_{\Delta I} \mathcal{L}_{CE}\left(g_\phi(I_m + \Delta I), y_i\right), \tag{6}$$

$$\text{s.t.} \quad \|\Delta I\|_p \leq \epsilon, \tag{7}$$

where $I$ is the to-be-manipulated image, $I_m$ is the variable to be optimized and represents the manipulation result. $\epsilon$ is the maximum allowable value of attacks, which we set to be $8/255$. The first term is to guarantee the quality of the manipulated image. The second term is to enhance the robustness of downstream models against attacks during creation and transmission.

We do not construct deep neural networks for data representation in this method. Similarly, we use Adam (Kingma & Ba, 2015) as the optimizer, with default parameters and a learning rate of $\mu = 0.01$. A total of 1000 optimization steps are performed, with each step involving the entire set of pixels of the to-be-manipulated image.

## B    MOTIVATION OF DPGD

In PGD attacks against images, the perturbation is projected to a specified pixel range in each iteration. However, attacks on INR do not directly manipulate the image. If only the fluctuation range of INR parameters is considered, the image decoded from the perturbed INR could be severely damaged. Therefore, we consider adding constraints directly on the image level for the gradients. The detailed analysis is as follows. The loss function is defined as $J(\theta) = \mathcal{L}_{CE}(g_\phi(\hat{I}), y)$. Here, $\hat{I} = \Psi(f_\theta)$ represents the image reconstructed by INR $f_\theta$. The original input's INR weight is $\theta_{org}$, and the image it reconstructs is $\hat{I}_{org}$. In the $t$-th iteration, the image reconstructed by the INR with weights $\theta_{t-1}$ before the attack is $\hat{I}_{t-1}$, and the loss is $J(\theta_{t-1})$. The gradient of the loss at $\theta_{t-1}$ is calculated as $\nabla_{\theta_{t-1}} = \frac{\partial J}{\partial \theta_{t-1}}$. Updating against the gradient direction, we get $\theta_t = \theta_{t-1} + \alpha \frac{\partial J}{\partial \theta_{t-1}}$. At this point, the new loss $J'$ can be estimated as:

$$
\begin{aligned}
J(\theta_t) &\approx J(\theta_{t-1}) + (\theta_t - \theta_{t-1})\frac{\partial J}{\partial \theta_{t-1}} \\
&= J(\theta_{t-1}) + \alpha \left( \frac{\partial J}{\partial \theta_{t-1}} \right)^2 .
\end{aligned}
\tag{8}
$$

Since the second term in the equation above is always positive, it allows for updates in a direction that increases the loss gradually. Now, we consider the impact of weight updates on the reconstructed image. The reconstructed image can be approximated as

$$
\begin{aligned}
\hat{I}_t &\approx \hat{I}_{t-1} + (\theta_t - \theta_{t-1})\frac{\partial \hat{I}}{\partial \theta_{t-1}} \\
&= \hat{I}_{t-1} + \alpha \frac{\partial J}{\partial \theta_{t-1}} \frac{\partial \hat{I}}{\partial \theta_{t-1}} \\
&= \hat{I}_{t-1} + \alpha \frac{\partial J}{\partial \hat{I}} \frac{\partial \hat{I}}{\partial \theta_{t-1}} \frac{\partial \hat{I}}{\partial \theta_{t-1}} \\
&= \hat{I}_{t-1} + \alpha \left( \frac{\partial \hat{I}}{\partial \theta_{t-1}} \right)^2 \frac{\partial J}{\partial \hat{I}} .
\end{aligned}
\tag{9}
$$

To prevent significant differences between the reconstructed image and the original image, we project $\hat{I}_t$ onto the $\ell_p$ ball around the original reconstructed image $\hat{I}_{org}$. Therefore, the following constraint is added:

$$
||\hat{I}_t - \hat{I}_{\text{org}}||_p \leq \zeta.
\tag{10}
$$

That is

$$
||\hat{I}_{t-1} + \alpha(\frac{\partial \hat{I}}{\partial \theta_{t-1}})^2 \frac{\partial J}{\partial \hat{I}} - \hat{I}_{\text{org}}||_p \leq \zeta.
\tag{11}
$$

Since $\alpha(\frac{\partial \hat{I}}{\partial \theta_{t-1}})^2$ is always positive, for ease of computation, we simplify it as $\alpha(\frac{\partial \hat{I}}{\partial \theta_{t-1}})^2 = 1$, which leads to line 7 in Algorithm 2.

## C    ADDITIONAL RESULTS

### C.1    ADDITIONAL RESULTS FOR OUR DEFENSE-IN-CREATION

We provide additional results for our defense-in-creation approach. In our experiment, we select PreActResNet-18 (He et al., 2016) and WideResNet34-10 (Zagoruyko & Komodakis, 2016) as the

downstream classifier architectures. The weight of reconstruction loss $\lambda_1$ in Equation 4 is fixed as $\lambda_1 = 1$, and the weight of reconstruction loss $\lambda_2$ varies to balance the reconstruction quality and robustness. For CIFAR-10, the results for PreActResNet-18 and WideResNet34-10 are presented in Table 4 and Table 5, respectively. For CIFAR-100, the results for PreActResNet-18 and WideResNet34-10 are presented in Table 6 and Table 7, respectively. For SVHN, the results for PreActResNet-18 and WideResNet34-10 are presented in Table 8. The case where the weight of reconstruction loss $\lambda_2 = 0$ in the results represents the normal INR optimization, considering only the reconstruction loss. Our findings demonstrate that our approach effectively strengthens the robustness of the downstream models built on the PreActResNet-18 and WideResNet34-10 architectures for all datasets.

### C.2 ADDITIONAL RESULTS FOR DIRECT PIXEL MANIPULATION

Direct pixel manipulation is to solve a min-max game defined in Equation 6. The hyper-parameter $\lambda_1$ and $\lambda_2$ in Equation 6 are used to balance the image quality and the robustness of downstream models. We present additional results showcasing direct pixel manipulation using various ratios of $\lambda_1$ and $\lambda_2$. The results are shown in Table 9. Although direct pixel manipulation can provide some defense against adversarial attacks, it exhibits significant image distortion and inferior defense efficacy compared to our defense-in-creation approach.

## D COMPARISON WITH IMAGE-BASED ADVERSARIAL ATTACK & DEFENSE

Our method is different from image-based adversarial attacks as we perform attacks on the parameters of INR. When evaluating defense capabilities, it is necessary to select appropriate parameters to ensure consistent attack strength. Due to the inconsistency between these two attack forms, it is a challenge to correlate the strength of these two attacks.

In this article, considering that adversarial samples should be imperceptible to humans, our approach is to use image quality as a criterion. Specifically, when conducting traditional image-based adversarial attacks, we select parameters widely used in existing literature (Szegedy et al., 2014; Madry et al., 2018; Zhang et al., 2019). We then evaluate the image quality after the attack. When attacking INR, we try to make the quality of the reconstructed image after the attack as close as possible to the image quality after the image-based attack. At this point, we can consider these two attacks, one targeting images and the other targeting INR, to be consistent. In the actual execution process, due to the inability to achieve complete accuracy, we make the reconstructed quality after the attack on INR worse, which indicates that our method will encounter greater attack intensity. The results can demonstrate the effectiveness of our proposed method.

Table 4: Evaluation of our defense-in-creation using different weights of robust loss $\lambda_2$. We evaluate the performance of the defense against attacks during transmission. The results are averaged on all examples in CIFAR-10 test dataset. The classifier architecture employed is PreActResNet-18.

| | $\lambda_2$ | 0 | 0.0002 | 0.0004 | 0.0006 | 0.0008 | 0.001 | 0.0012 | 0.0014 | 0.0016 |
|---|---|---|---|---|---|---|---|---|---|---|
| Natural | Accuracy | 94.73% | 100% | 100% | 100% | 100% | 100% | 100% | 100% | 100% |
| | PSNR | 48.384 | 45.832 | 45.194 | 44.766 | 44.395 | 44.132 | 43.907 | 43.666 | 43.489 |
| | LPIPS | 0.002 | 0.004 | 0.005 | 0.006 | 0.007 | 0.007 | 0.008 | 0.008 | 0.009 |
| | SSIM | 0.998 | 0.997 | 0.996 | 0.996 | 0.995 | 0.995 | 0.995 | 0.994 | 0.994 |
| DPGD | Accuracy | 20.80% | 97.84% | 98.56% | 98.84% | 99.40% | 99.32% | 99.28% | 99.32% | 99.28% |
| | PSNR | 28.551 | 28.122 | 28.201 | 28.313 | 28.287 | 28.364 | 28.374 | 28.414 | 28.502 |
| | LPIPS | 0.060 | 0.053 | 0.051 | 0.051 | 0.051 | 0.050 | 0.049 | 0.049 | 0.049 |
| | SSIM | 0.975 | 0.979 | 0.979 | 0.978 | 0.978 | 0.978 | 0.978 | 0.977 | 0.977 |
| FGSM for INR | Accuracy | 57.36% | 100% | 100% | 100% | 100% | 100% | 100% | 100% | 100% |
| | PSNR | 24.564 | 25.501 | 25.651 | 25.708 | 25.766 | 25.775 | 25.877 | 25.918 | 25.946 |
| | LPIPS | 0.111 | 0.086 | 0.083 | 0.082 | 0.080 | 0.080 | 0.077 | 0.077 | 0.076 |
| | SSIM | 0.959 | 0.966 | 0.966 | 0.965 | 0.965 | 0.964 | 0.965 | 0.964 | 0.963 |
| PGD for INR | Accuracy | 18.84% | 93.96% | 96.12% | 97.44% | 97.72% | 97.76% | 98.60% | 98.60% | 98.38% |
| | PSNR | 26.474 | 27.919 | 28.050 | 28.221 | 28.246 | 28.260 | 28.362 | 28.400 | 28.433 |
| | LPIPS | 0.077 | 0.054 | 0.051 | 0.050 | 0.050 | 0.050 | 0.049 | 0.049 | 0.049 |
| | SSIM | 0.970 | 0.979 | 0.978 | 0.978 | 0.978 | 0.978 | 0.978 | 0.977 | 0.977 |
| CW for INR | Accuracy | 17.72% | 52.24% | 55.64% | 56.88% | 58.12% | 58.72% | 58.68% | 58.36% | 59.44% |
| | PSNR | 26.622 | 27.157 | 27.276 | 27.262 | 27.314 | 27.298 | 27.332 | 27.419 | 27.437 |
| | LPIPS | 0.075 | 0.067 | 0.065 | 0.064 | 0.064 | 0.065 | 0.064 | 0.063 | 0.063 |
| | SSIM | 0.970 | 0.975 | 0.975 | 0.974 | 0.974 | 0.973 | 0.973 | 0.973 | 0.972 |

| | $\lambda_2$ | 0.0018 | 0.002 | 0.0022 | 0.0024 | 0.0026 | 0.0028 | 0.003 | 0.0032 |
|---|---|---|---|---|---|---|---|---|---|
| Natural | Accuracy | 100% | 100% | 100% | 100% | 100% | 100% | 100% | 100% |
| | PSNR | 43.251 | 43.146 | 42.920 | 42.750 | 42.662 | 42.443 | 42.265 | 42.194 |
| | LPIPS | 0.009 | 0.010 | 0.010 | 0.011 | 0.011 | 0.012 | 0.013 | 0.013 |
| | SSIM | 0.994 | 0.994 | 0.993 | 0.993 | 0.992 | 0.992 | 0.992 | 0.992 |
| DPGD | Accuracy | 100% | 99% | 100% | 99% | 99% | 100% | 100% | 100% |
| | PSNR | 28.457 | 28.537 | 28.539 | 28.526 | 28.600 | 28.652 | 28.565 | 28.590 |
| | LPIPS | 0.048 | 0.048 | 0.049 | 0.049 | 0.049 | 0.049 | 0.050 | 0.049 |
| | SSIM | 0.977 | 0.977 | 0.976 | 0.976 | 0.976 | 0.975 | 0.975 | 0.975 |
| FGSM for INR | Accuracy | 100% | 100% | 100% | 100% | 100% | 100% | 100% | 100% |
| | PSNR | 25.942 | 26.001 | 25.967 | 25.995 | 26.059 | 26.059 | 26.058 | 26.033 |
| | LPIPS | 0.075 | 0.075 | 0.075 | 0.075 | 0.074 | 0.074 | 0.075 | 0.074 |
| | SSIM | 0.963 | 0.963 | 0.963 | 0.962 | 0.962 | 0.961 | 0.961 | 0.960 |
| PGD for INR | Accuracy | 98% | 98% | 99% | 99% | 99% | 99% | 99% | 99% |
| | PSNR | 28.450 | 28.473 | 28.482 | 28.490 | 28.560 | 28.639 | 28.555 | 28.586 |
| | LPIPS | 0.048 | 0.049 | 0.049 | 0.050 | 0.049 | 0.049 | 0.050 | 0.049 |
| | SSIM | 0.977 | 0.977 | 0.976 | 0.976 | 0.976 | 0.975 | 0.975 | 0.975 |
| CW for INR | Accuracy | 59% | 60% | 61% | 60% | 60% | 62% | 60% | 61% |
| | PSNR | 27.459 | 27.473 | 27.468 | 27.474 | 27.502 | 27.514 | 27.482 | 27.528 |
| | LPIPS | 0.062 | 0.062 | 0.062 | 0.063 | 0.063 | 0.063 | 0.063 | 0.063 |
| | SSIM | 0.972 | 0.972 | 0.971 | 0.971 | 0.970 | 0.970 | 0.970 | 0.970 |

Table 5: Evaluation of our defense-in-creation using different weights of robust loss $\lambda_2$. We evaluate the performance of the defense against attacks during transmission. The results are averaged on all examples in CIFAR-10 test dataset. The classifier architecture employed is WideResNet34-10.

|  | $\lambda_2$ | 0 | 0.0006 | 0.001 | 0.002 |
|---|---|---|---|---|---|
| Natural | Accuracy | 94.56% | 100.00% | 100.00% | 100.00% |
| | PSNR | 48.430 | 44.509 | 43.794 | 42.634 |
| | LPIPS | 0.002 | 0.006 | 0.008 | 0.011 |
| | SSIM | 0.998 | 0.995 | 0.994 | 0.993 |
| DPGD | Accuracy | 17.12% | 98.60% | 98.92% | 98.92% |
| | PSNR | 27.578 | 27.780 | 27.867 | 27.965 |
| | LPIPS | 0.075 | 0.059 | 0.059 | 0.057 |
| | SSIM | 0.968 | 0.975 | 0.974 | 0.973 |
| FGSM for INR | Accuracy | 55.92% | 100.00% | 100.00% | 100.00% |
| | PSNR | 24.762 | 25.855 | 25.942 | 26.028 |
| | LPIPS | 0.113 | 0.085 | 0.082 | 0.079 |
| | SSIM | 0.957 | 0.964 | 0.962 | 0.960 |
| PGD for INR | Accuracy | 15.28% | 95.80% | 97.20% | 98.08% |
| | PSNR | 25.952 | 27.727 | 27.790 | 27.939 |
| | LPIPS | 0.090 | 0.060 | 0.059 | 0.057 |
| | SSIM | 0.964 | 0.975 | 0.974 | 0.972 |
| CW for INR | Accuracy | 12.80% | 43.48% | 43.68% | 42.20% |
| | PSNR | 26.041 | 26.803 | 26.859 | 26.806 |
| | LPIPS | 0.087 | 0.074 | 0.073 | 0.074 |
| | SSIM | 0.964 | 0.969 | 0.968 | 0.965 |

Table 6: Evaluation of our defense-in-creation using different weights of robust loss $\lambda_2$. We evaluate the performance of the defense against attacks during transmission. The results are averaged on all examples in CIFAR-100 test dataset. The classifier architecture employed is PreActResNet-18.

| | $\lambda_2$ | 0 | 0.0002 | 0.0004 | 0.0006 | 0.0008 | 0.001 | 0.0012 | 0.0014 | 0.0016 |
|---|---|---|---|---|---|---|---|---|---|---|
| Natural | Accuracy | 76.44% | 100.00% | 100.00% | 100.00% | 100.00% | 100.00% | 100.00% | 100.00% | 100.00% |
| | PSNR | 48.265 | 42.595 | 41.344 | 40.498 | 39.761 | 39.165 | 38.608 | 38.161 | 37.762 |
| | LPIPS | 0.002 | 0.011 | 0.015 | 0.018 | 0.022 | 0.025 | 0.028 | 0.031 | 0.034 |
| | SSIM | 0.998 | 0.992 | 0.989 | 0.987 | 0.985 | 0.982 | 0.980 | 0.978 | 0.976 |
| DPGD | Accuracy | 4.32% | 98.32% | 98.96% | 98.96% | 99.40% | 99.44% | 99.56% | 99.32% | 99.56% |
| | PSNR | 30.065 | 28.228 | 28.015 | 28.112 | 27.953 | 28.126 | 28.069 | 28.062 | 28.232 |
| | LPIPS | 0.044 | 0.050 | 0.051 | 0.052 | 0.055 | 0.055 | 0.058 | 0.059 | 0.061 |
| | SSIM | 0.974 | 0.971 | 0.969 | 0.966 | 0.963 | 0.962 | 0.960 | 0.958 | 0.957 |
| FGSM for INR | Accuracy | 17.68% | 100.00% | 100.00% | 100.00% | 100.00% | 100.00% | 100.00% | 100.00% | 100.00% |
| | PSNR | 24.508 | 25.857 | 25.945 | 25.974 | 26.024 | 26.027 | 26.066 | 26.152 | 26.059 |
| | LPIPS | 0.107 | 0.081 | 0.080 | 0.080 | 0.081 | 0.082 | 0.084 | 0.084 | 0.086 |
| | SSIM | 0.952 | 0.961 | 0.959 | 0.957 | 0.954 | 0.952 | 0.951 | 0.949 | 0.947 |
| PGD for INR | Accuracy | 2.16% | 95.84% | 98.24% | 98.44% | 98.60% | 99.20% | 98.84% | 98.68% | 99.20% |
| | PSNR | 26.560 | 27.647 | 27.604 | 27.738 | 27.699 | 27.716 | 27.791 | 27.738 | 27.805 |
| | LPIPS | 0.070 | 0.055 | 0.056 | 0.057 | 0.060 | 0.061 | 0.064 | 0.065 | 0.067 |
| | SSIM | 0.966 | 0.971 | 0.969 | 0.968 | 0.965 | 0.964 | 0.962 | 0.960 | 0.958 |
| CW for INR | Accuracy | 1.76% | 67.32% | 73.24% | 74.52% | 75.00% | 75.24% | 75.56% | 76.60% | 76.12% |
| | PSNR | 26.734 | 27.189 | 27.193 | 27.143 | 27.232 | 27.245 | 27.288 | 27.252 | 27.241 |
| | LPIPS | 0.067 | 0.062 | 0.063 | 0.065 | 0.066 | 0.068 | 0.070 | 0.072 | 0.073 |
| | SSIM | 0.967 | 0.969 | 0.967 | 0.965 | 0.963 | 0.961 | 0.959 | 0.957 | 0.955 |

| | $\lambda_2$ | 0.0018 | 0.002 | 0.0022 | 0.0024 | 0.0026 | 0.0028 | 0.003 | 0.0032 |
|---|---|---|---|---|---|---|---|---|---|
| Natural | Accuracy | 100% | 100% | 100% | 100% | 100% | 100% | 100% | 100% |
| | PSNR | 37.389 | 37.010 | 36.744 | 36.457 | 36.147 | 35.916 | 35.622 | 35.450 |
| | LPIPS | 0.037 | 0.039 | 0.042 | 0.044 | 0.047 | 0.050 | 0.052 | 0.054 |
| | SSIM | 0.974 | 0.972 | 0.970 | 0.968 | 0.966 | 0.964 | 0.962 | 0.960 |
| DPGD | Accuracy | 99.08% | 99.36% | 99.48% | 99.52% | 99.72% | 99.76% | 99.68% | 99.72% |
| | PSNR | 28.044 | 28.009 | 27.944 | 28.026 | 27.966 | 27.904 | 27.865 | 27.803 |
| | LPIPS | 0.063 | 0.065 | 0.067 | 0.068 | 0.071 | 0.073 | 0.075 | 0.076 |
| | SSIM | 0.954 | 0.953 | 0.951 | 0.950 | 0.948 | 0.946 | 0.944 | 0.943 |
| FGSM for INR | Accuracy | 100.00% | 100.00% | 100.00% | 100.00% | 100.00% | 100.00% | 100.00% | 100.00% |
| | PSNR | 26.223 | 26.061 | 26.157 | 26.152 | 26.091 | 26.128 | 26.107 | 26.189 |
| | LPIPS | 0.089 | 0.090 | 0.091 | 0.093 | 0.095 | 0.095 | 0.098 | 0.099 |
| | SSIM | 0.945 | 0.943 | 0.942 | 0.940 | 0.938 | 0.937 | 0.934 | 0.933 |
| PGD for INR | Accuracy | 98.88% | 98.92% | 99.04% | 99.12% | 99.08% | 99.28% | 99.24% | 99.16% |
| | PSNR | 27.769 | 27.697 | 27.731 | 27.712 | 27.662 | 27.674 | 27.649 | 27.635 |
| | LPIPS | 0.070 | 0.072 | 0.074 | 0.075 | 0.077 | 0.079 | 0.081 | 0.083 |
| | SSIM | 0.956 | 0.954 | 0.953 | 0.951 | 0.949 | 0.948 | 0.945 | 0.944 |
| CW for INR | Accuracy | 76.64% | 77.20% | 75.60% | 77.12% | 76.88% | 76.60% | 78.64% | 76.96% |
| | PSNR | 27.225 | 27.185 | 27.190 | 27.201 | 27.181 | 27.163 | 27.176 | 27.197 |
| | LPIPS | 0.075 | 0.078 | 0.079 | 0.081 | 0.084 | 0.086 | 0.087 | 0.089 |
| | SSIM | 0.953 | 0.951 | 0.949 | 0.948 | 0.946 | 0.944 | 0.943 | 0.941 |

Table 7: Evaluation of our defense-in-creation using different weights of robust loss $\lambda_2$. We evaluate the performance of the defense against attacks during transmission. The results are averaged on all examples in CIFAR-100 test dataset. The classifier architecture employed is WideResNet34-10.

| | $\lambda_2$ | 0 | 0.001 | 0.002 |
|---|---|---|---|---|
| Natural | Accuracy | 80.00% | 100.00% | 100.00% |
| | PSNR | 48.340 | 40.087 | 37.847 |
| | LPIPS | 0.002 | 0.020 | 0.033 |
| | SSIM | 0.998 | 0.985 | 0.975 |
| DPGD | Accuracy | 8.56% | 99.40% | 99.32% |
| | PSNR | 29.433 | 28.022 | 27.876 |
| | LPIPS | 0.045 | 0.057 | 0.067 |
| | SSIM | 0.974 | 0.967 | 0.958 |
| FGSM for INR | Accuracy | 33.68% | 100.00% | 100.00% |
| | PSNR | 24.912 | 26.525 | 26.455 |
| | LPIPS | 0.098 | 0.070 | 0.077 |
| | SSIM | 0.955 | 0.956 | 0.947 |
| PGD for INR | Accuracy | 7.44% | 98.56% | 98.40% |
| | PSNR | 26.824 | 27.955 | 27.818 |
| | LPIPS | 0.067 | 0.057 | 0.067 |
| | SSIM | 0.969 | 0.967 | 0.958 |
| CW for INR | Accuracy | 5.88% | 46.44% | 51.04% |
| | PSNR | 26.986 | 27.000 | 26.920 |
| | LPIPS | 0.065 | 0.070 | 0.077 |
| | SSIM | 0.970 | 0.961 | 0.952 |

Table 8: Evaluation of our defense-in-creation using different weights of robust loss $\lambda_2$. We evaluate the performance of the defense against attacks during transmission. The results are averaged on all examples in SVHN test dataset. The classifier architectures are indicated in the table.

| PreActResNet-18 | | | | | | WideResNet34-10 | | | | | |
|---|---|---|---|---|---|---|---|---|---|---|---|
| | $\lambda_2$ | 0 | 0.0006 | 0.001 | 0.002 | | $\lambda_2$ | 0 | 0.001 | 0.002 | 0.003 |
| Natural | Accuracy | 95.64% | 100.00% | 100.00% | 100.00% | Natural | Accuracy | 95.80% | 100.00% | 100.00% | 100.00% |
| | PSNR | 53.004 | 47.722 | 46.785 | 45.229 | | PSNR | 53.032 | 46.985 | 45.676 | 44.748 |
| | LPIPS | 0.002 | 0.010 | 0.013 | 0.018 | | LPIPS | 0.002 | 0.012 | 0.016 | 0.019 |
| | SSIM | 0.999 | 0.995 | 0.993 | 0.990 | | SSIM | 0.999 | 0.994 | 0.992 | 0.990 |
| DPGD | Accuracy | 37.40% | 99.80% | 99.88% | 99.96% | DPGD | Accuracy | 39.04% | 99.88% | 100.00% | 99.96% |
| | PSNR | 28.541 | 29.297 | 29.396 | 29.717 | | PSNR | 27.401 | 28.427 | 28.736 | 28.846 |
| | LPIPS | 0.072 | 0.078 | 0.076 | 0.076 | | LPIPS | 0.089 | 0.103 | 0.101 | 0.101 |
| | SSIM | 0.972 | 0.978 | 0.977 | 0.975 | | SSIM | 0.967 | 0.977 | 0.976 | 0.975 |
| FGSM for INR | Accuracy | 66.44% | 100.00% | 100.00% | 100.00% | FGSM for INR | Accuracy | 65.96% | 100.00% | 100.00% | 100.00% |
| | PSNR | 27.291 | 27.675 | 27.859 | 28.160 | | PSNR | 27.254 | 27.829 | 28.139 | 28.254 |
| | LPIPS | 0.128 | 0.115 | 0.112 | 0.108 | | LPIPS | 0.133 | 0.120 | 0.116 | 0.113 |
| | SSIM | 0.965 | 0.970 | 0.969 | 0.966 | | SSIM | 0.964 | 0.973 | 0.972 | 0.971 |
| PGD for INR | Accuracy | 34.88% | 99.48% | 99.84% | 99.72% | PGD for INR | Accuracy | 36.24% | 99.88% | 99.92% | 99.92% |
| | PSNR | 29.032 | 29.299 | 29.415 | 29.726 | | PSNR | 28.443 | 28.429 | 28.708 | 28.871 |
| | LPIPS | 0.086 | 0.078 | 0.077 | 0.077 | | LPIPS | 0.103 | 0.103 | 0.101 | 0.100 |
| | SSIM | 0.973 | 0.978 | 0.977 | 0.975 | | SSIM | 0.969 | 0.977 | 0.976 | 0.975 |
| CW for INR | Accuracy | 35.52% | 82.72% | 84.80% | 83.76% | CW for INR | Accuracy | 35.76% | 84.84% | 84.68% | 85.16% |
| | PSNR | 29.187 | 29.144 | 29.193 | 29.432 | | PSNR | 28.572 | 28.564 | 28.646 | 28.715 |
| | LPIPS | 0.084 | 0.084 | 0.083 | 0.084 | | LPIPS | 0.100 | 0.100 | 0.100 | 0.100 |
| | SSIM | 0.974 | 0.976 | 0.974 | 0.971 | | SSIM | 0.970 | 0.973 | 0.970 | 0.968 |

Table 9: Evaluation of direct pixel manipulation using different weights of robust loss $\lambda_2$. We evaluate the performance of the defense against attacks during transmission. The results are averaged on all examples in SVHN test dataset. The classifier architectures are indicated in the table.

| PreActResNet-18 | | | | | WideResNet34-10 | | | |
|---|---|---|---|---|---|---|---|---|
| | $\lambda_2$ | 0.0006 | 0.001 | | | $\lambda_2$ | 0.0006 | 0.001 |
| Natural | Accuracy | 100.00% | 100.00% | | Natural | Accuracy | 100.00% | 100.00% |
| | PSNR | 34.498 | 33.395 | | | PSNR | 35.780 | 34.771 |
| | LPIPS | 0.185 | 0.209 | | | LPIPS | 0.148 | 0.170 |
| | SSIM | 0.895 | 0.874 | | | SSIM | 0.918 | 0.903 |
| FGSM | Accuracy | 100.00% | 100.00% | | FGSM | Accuracy | 100.00% | 100.00% |
| | PSNR | 29.061 | 28.685 | | | PSNR | 29.454 | 29.238 |
| | LPIPS | 0.297 | 0.307 | | | LPIPS | 0.265 | 0.272 |
| | SSIM | 0.808 | 0.794 | | | SSIM | 0.823 | 0.814 |
| PGD | Accuracy | 57.24% | 72.36% | | PGD | Accuracy | 76.60% | 87.84% |
| | PSNR | 30.830 | 30.348 | | | PSNR | 31.181 | 30.909 |
| | LPIPS | 0.252 | 0.265 | | | LPIPS | 0.231 | 0.239 |
| | SSIM | 0.857 | 0.842 | | | SSIM | 0.871 | 0.861 |
| CW | Accuracy | 26.24% | 35.48% | | CW | Accuracy | 18.24% | 27.48% |
| | PSNR | 30.673 | 30.195 | | | PSNR | 30.942 | 30.655 |
| | LPIPS | 0.264 | 0.275 | | | LPIPS | 0.246 | 0.254 |
| | SSIM | 0.852 | 0.836 | | | SSIM | 0.865 | 0.856 |

