# OpenReview forum: "Adversarial Data Robustness via Implicit Neural Representation"
_ICLR.cc/2024/Conference — Submitted to ICLR 2024_

### Official Review · Reviewer_GzWz · 2023-10-29

**Soundness:** 3 good
**Presentation:** 3 good
**Contribution:** 3 good
**Rating:** 3
**Confidence:** 3

**Summary:**

Considering that adversarial training for models are sometimes difficult for common users, the paper proposes a data-level robustness method called Implicit Neural Representation (INR) for data-level adversarial training, which adversarially trains INR to get robust data features without losing their contents.

**Strengths:**

1. The adversarial data robustness proposed in this paper is original, since most of the previous works focused on models' robustness, while the abstacles in their applications were hardly considered.

2. The idea of double projection innovatively explored the functions of projection in gradient-based attacks.

**Weaknesses:**

1. Since data-level robustness is a new area, the meaning about robustness designed with spatial coordinates is not clear.

2. The explanation about why the robust data can only be created via adversarial training in Section 3.1 is not convincing enough.

**Questions:**

1. Since data-level robustness is a new area, the meaning about robustness designed with spatial coordinates is not clear.

2. The explanation about why the robust data can only be created via adversarial training in Section 3.1 is not convincing enough.

Update after discussion

After discussion, the other reviewers and I think this work has a serious flaw in the use of label information. Thus, I will revise the rating score to 3.

---

> ### Author Response · Authors · 2023-11-21
> **Response to reviewer GzWz**
>
> > Since data-level robustness is a new area, the meaning about robustness designed with spatial coordinates is not clear.
>
> **Response:** Robustness is not directly related to spatial coordinates. The spatial coordinates are used as the input of the INR to output the corresponding RGB values. Our method allows for the incorporation of the classifier's relevant information into the parameters of the INR to force the INR to reach a robust state.
>
> > The explanation about why the robust data can only be created via adversarial training in Section 3.1 is not convincing enough.
>
> **Response:** Thank you for your suggestion. We think adversarial training could be an effective method for robust data generation. So, we change this sentence into:
>
> "Considering the effectiveness of adversarial training in countering adversarial attacks, it could also be a valuable approach for generating robust data."

---

### Official Review · Reviewer_7WdV · 2023-10-29

**Soundness:** 3 good
**Presentation:** 3 good
**Contribution:** 3 good
**Rating:** 8
**Confidence:** 4

**Summary:**

This paper investigates adversarial data robustness by applying Implicit Neural Representation (INR), storing the image as a network-based representation, and then reconstructing the representation with robustness enhancement before calling task models. With adversarial data robustness, the users do not need to worry about the model-level robustness with adversarial training in practice. The paper looked into two different attack stages and proposed adaptive attack DPGD. Last, the paper proposed a new defense-in-creation strategy for defense and extensively evaluate the performance.

**Strengths:**

The paper proposes a novel direction for adversarial robustness on data preparation stages. It prevents user enhancing model robustness with adversarial training while still achieving good robustness behavior.

According to the empirical experiments, the strategies achieve promising results compared to traditional model-level robust training.

The paper is well-written and easy to follow.

**Weaknesses:**

The authors motivate the adversarial data robustness by assuming many users do not have knowledge for adversarial training on their models. However, the proposed strategy still needs robust training for INR model. This is one caveat for the motivation.

It would be helpful to evaluate stronger attacks like Auto Attack besides CW and PGD.

**Questions:**

Is this approach compatible to model-level adversarial training? I am curious if applying both data-level and model-level approaches would lead to stronger robustness over SOTA?

---

> ### Author Response · Authors · 2023-11-21
> **Response to reviewer 7WdV**
>
> > The authors motivate the adversarial data robustness by assuming many users do not have knowledge for adversarial training on their models. However, the proposed strategy still needs robust training for INR model. This is one caveat for the motivation.
>
> Our approach is particularly suitable for situations where adversarial training at the model level is challenging. For instance, adversarially training a large-scale foundational model can be complex. In response, model developers can define a secure data format within their frameworks to enhance safety. Additionally, our concept of 'defense-in-creation' could serve as a safety module integrated with the foundational model. This setup would require that any images inputted into such a large-scale foundational model first pass through this safety module, which would remove adversarial perturbations.
>
> > It would be helpful to evaluate stronger attacks like Auto Attack besides CW and PGD.
>
> Thank you for your suggestion. We conducted experiments by adapting AutoAttack for images during INR creation, and the results show that our defense-in-creation can reach high accuracy. We will add the results to the revised paper. For attack during transmission, due to the data being represented as INR, it is not feasible to directly apply AutoAttack to the INR parameters. Thus, the current AutoAttack cannot directly attack INR parameters.
>
> > Is this approach compatible with model-level adversarial training? I am curious if applying both data-level and model-level approaches would lead to stronger robustness over SOTA?
>
> Yes. It is compatible with model-level adversarial training. As we do not have specific requirements for downstream models, they can also be an adversarial-robust network model. We will further study the cooperation with model-level approaches in our future work.

---

### Official Review · Reviewer_bc25 · 2023-10-30

**Soundness:** 1 poor
**Presentation:** 2 fair
**Contribution:** 1 poor
**Rating:** 3
**Confidence:** 4

**Summary:**

This paper proposes a method that makes data resistant to adversarial perturbations, ensuring that downstream models remain robust even when subjected to attacks. The author achieves this by representing data as Implicit Neural Representations (INRs), without the need to modify the deep learning models’ weights.

**Strengths:**

The proposed concept of *adversarial data robustness* is interesting and leveraging the INR framework to achieve data robustness is novel.

**Weaknesses:**

This paper is subject to several weaknesses that need to be addressed.

1. **(Motivation)** The paper's motivation for storing data as a learnable representation via INR in the context of adversarial robustness is not well-established. Additionally, the proposed attack scenario, involving direct manipulation of INR parameters during data transmission, raises concerns about its practicality and real-world relevance. It would be beneficial for the authors to provide a more robust justification for these choices and consider the feasibility of such attacks in practical applications to ensure a more realistic context for the proposed defense strategy.

2. **(Method)** The paper introduces the concept of storing data as INRs for adversarial robustness, but it lacks clarity on how these representations can be derived for testing data. Given that testing data is typically only accessible during the testing phase, the paper should address the challenge of generating specific INRs for each testing image. For now, I would treat INR as a mere image generator.

3. **(Experiment)** The experimental design in the paper appears to lack fairness, which fails to prove the efficacy of the proposed method. Also, it is unclear if the proposed framework could be used to defend against unseen attacks (e.g., [1]).


[1] Perceptual adversarial robustness: Defense against unseen threat models. (ICLR 2021)

**Questions:**

1. Please provide a more in-depth justification for the choice of INR as a data representation method for adversarial robustness.

2. How do the authors envision real-world applications where adversarial data robustness using INR would be particularly beneficial?

3. In the proposed setup, how can users derive INR representations for testing data, considering that in adversarial training, testing data is typically not accessible during training?

4. Are there any potential biases or confounding factors in the experimental setup that need to be addressed?

5. Can the proposed framework used to defend unseen attacks?

6. Please provide the computational cost of the proposed method and compare it with other baselines.

---

> ### Author Response · Authors · 2023-11-21
> **Response to reviewer bc25**
>
> > The paper introduces the concept of storing data as INRs for adversarial robustness, but it lacks clarity on how these representations can be derived for testing data. Given that testing data is typically only accessible during the testing phase, the paper should address the challenge of generating specific INRs for each testing image. For now, I would treat INR as a mere image generator.
>
> **Response:** As we said in our submission, even for a new image, we only need to represent it to its corresponding INR format by incorporating the classifier property. Thus, in our experiments, for each classifier, we directly transfer their corresponding testing data into the INR format by incorporating the classifier property. Then, we can achieve the envisioned robustness.
>
> > The experimental design in the paper appears to lack fairness, which fails to prove the efficacy of the proposed method.
>
> **Response:** We propose to incorporate the classifier's related information into the data format. Therefore, we should train INR for each image individually. We spend time on INR creation, but not on training models adversarially. Therefore, it is hard to compare the efficacy with other methods simply.
>
> > Please provide a more in-depth justification for the choice of INR as a data representation method for adversarial robustness.
>
> **Response:**  We just utilize the distinguished overfitting property of INR [1] to incorporate the classifier property into the data format. Then, we can achieve higher robustness.  We can also consider image pixels as learnable parameters and optimize these learnable parameters adversarially. However, such a straightforward solution significantly undermines image quality since the pixel values are directly manipulated.
>
> [1] Strümpler, Yannick, et al. "Implicit neural representations for image compression." ECCV. 2022.
>
> > How do the authors envision real-world applications where adversarial data robustness using INR would be particularly beneficial?
>
> **Response:**  This is needed when the model developers have a higher demand for robustness especially for some established specific models. In this situation, if they have high demands for the safety of their models and feel reluctant to do any alterations to their models, our setup would be useful for them.  They only need to create robust data without any alternations to their models.
>
> > In the proposed setup, how can users derive INR representations for testing data, considering that in adversarial training, testing data is typically not accessible during training?
>
> **Response**: Even for a totally new image, we only need to overfit it to its corresponding INR format by considering the classifier property. Thus, in our experiments, for each classifier, we directly transfer their corresponding testing data into the INR format by incorporating the classifier property. Then, we can achieve the envisioned robustness.
>
> > Are there any potential biases or confounding factors in the experimental setup that need to be addressed?
>
> **Response**: We have emphasized in the article that due to the distinct nature of the envisioned scenario from traditional adversarial defense, our experimental setup differs accordingly. We obtain robust data representations for each image through defense-in-creation, which can withstand attacks during both creation and transmission. Similar to other INR methods, we do not specifically require a training dataset, which sets it apart from the traditional scenario.
>
> > Can the proposed framework used to defend unseen attacks?
>
> Yes. In our experiment setting, we use 10-step PGD for INR training, while we test FGSM, 100-step PGD, and CW attacks for evaluation. We present the results of the experiments in Table 2.
>
>
> > Please provide the computational cost of the proposed method and compare it with other baselines.
>
> In our experiments, we did not focus on time efficiency or computational costs, as most of the tasks could be completed swiftly using a single GPU. Typically, standard INR encoding is accomplished in about 7.12 seconds. When incorporating data robustness into the process, the entire training session can be concluded in under 175.50 seconds. Since conventional adversarial training methods like PGD-AT take 3-30 times longer than standard training [1], the time consumption of our method is also acceptable. It is also worth noting from prior research that 7-PGD adversarial training for CIFAR-10 takes more than 5000 minutes [2]. Besides, adversarial training for a diffusion model may require "millions of generated data even on small datasets such as CIFAR-10 on 8 $\times$ A100 GPUs, which is inefficient in the training phase"[3].
>
>
> [1] Bai, Tao, et al. "Recent Advances in Adversarial Training for Adversarial Robustness." IJCAI 2021
>
> [2] Shafahi, Ali, et al. "Adversarial training for free!." NeurIPS 2019.
>
> [3] Wang, Zekai, et al. "Better diffusion models further improve adversarial training." ICML 2023.

---

> > ### Comment · Reviewer_bc25 · 2023-11-23
> >
> > Thanks for the response from the authors. However, my primary concern remains unaddressed.
> >
> > Labeling during the testing phase is *de facto* the problem. As I stated in my review: "*Given that testing data is typically only accessible during the testing phase, the paper should address the challenge of generating specific INRs for each testing image.*" It is not fair *to use the labels from the test set to generate robust data and study its robustness*.
> >
> > It is also weird why the INRs of the testing data should be released by the model developer, the reviewer is still struggling to come up with a scenario where the proposed work could be applied.

---

> > > ### Author Response · Authors · 2023-11-23
> > > **Response to reviewer bc25**
> > >
> > > Thank you for your comments.
> > >
> > > We know that our setting may make you feel weird, as data robustness is seldom considered before. However, our target is to generate robust data for downstream tasks, and such data can finally be a robust dataset for various purposes. A typical scenario is image retrieval. As the images in the gallery database are potentially to be attacked adversarially, our method can be used for database generation. The system developers can use our method to generate robust data that can be stored in the database for image retrieval purposes. The labels used here are the features obtained by the optimized feature encoder [1]. Then, if malicious attackers pollute or replace some samples in the database, our framework can still function properly. Actually, as we said in our manuscript in the last paragraph of Sec 4.3, even without the label, our method can still function properly.
> > >
> > > [1] An, Xiang, et al. "Unicom: Universal and Compact Representation Learning for Image Retrieval." ICLR. 2023.

---

### Official Review · Reviewer_Gy4W · 2023-10-31

**Soundness:** 2 fair
**Presentation:** 3 good
**Contribution:** 2 fair
**Rating:** 3
**Confidence:** 4

**Summary:**

This paper proposes adversarial data robustness, aiming to allow the data to resist adversarial perturbations. The proposed method stores the data as a learnable representation via Implicit Neural Representation (INR), and trains such a representation adversarially to achieve data robustness. Empirical evaluations are done on CIFAR-10/100 and SVHN, against PGD/DPGD, FGSM, and CW attacks.

**Strengths:**

The strengths of this paper include:
- The writing is clear with intuitive explanations such as Figures 1 and 3. The formulas in Section 3 is straightforward and easy to follow.
- I like the high-level idea of data robustness, which is supposed to be (conceptually) more efficient than model robustness.

**Weaknesses:**

The weaknesses of this paper include:
- The attack during creation formulated in Eq (2) is not an *adaptive attack* [1,2]. Specifically, an adaptive attacking objective should be like
$$\\max\_{\\Delta I\_{i}}\\mathcal{L}\_{CE}(g\_{\\phi}(\\Psi\\circ f\_{\theta}(I\_{i}+\\Delta I\_{i})),y\_{i})\\textrm{,}$$
where the INR decoding and reconstruction process implemented by $\Psi\circ f\_{\theta}$ should be involved.

- There are two extra computations introduced by the proposed method: first is the optimization process of $f\_{\theta}$, which is required to be optimized for each input image (i.e., cannot be amortized); second is the defense-in-creation process of Eq (4), which requires adversarial training by perturbing model parameters $\\theta$. The authors should report the empirical cost (e.g., computational time) for these two operations.

- The considered attacking methods such as PGD/DPGD, CW and FGSM are not strong. The authors should evaluate their methods under strong attacks like AutoAttack[3] and compare with the state-of-the-art models listed on RobustBench[4].


References: \
[1] Athalye et al. Obfuscated gradients give a false sense of security: Circumventing defenses to adversarial examples. ICML 2018 \
[2] Carlini et al. On evaluating adversarial robustness. arXiv 2019 \
[3] Croce and Hein. Reliable evaluation of adversarial robustness with an ensemble of diverse parameter-free attacks. ICML 2020 \
[4] https://robustbench.github.io/

**Questions:**

From my experience, the proposed method (with 92.51% accuracy against PGD) probably has gradient obfuscation [1,2]. The authors should evaluate their method under adaptive attacks that involving the defense mechanism (i.e., INR mechanism), as well as strong off-the-shelf attacks such as AutoAttack.


References: \
[1] Athalye et al. Obfuscated gradients give a false sense of security: Circumventing defenses to adversarial examples. ICML 2018 \
[2] Carlini et al. On evaluating adversarial robustness. arXiv 2019

---

> ### Author Response · Authors · 2023-11-21
> **Response to reviewer Gy4W - Part 1**
>
> > The attack during creation formulated in Eq (2) is not an *adaptive attack*. Specifically, an adaptive attacking objective should be like
> $$
> \max\_{\Delta I\_{i}}\mathcal{L}\_{CE}(g\_{\phi}(\Psi\circ f\_{\theta}(I\_{i}+\Delta I\_{i})),y\_{i})\textrm{,}
> $$
> where the INR decoding and reconstruction process implemented by $\Psi\circ f\_{\theta}$ should be involved.
>
> **Response:** We elucidate the connection between Eq. (2) and the adaptive attack. We define $\Phi: I \mapsto f\_\theta$ as the mapping from the image $I$ to the intermediate representation (INR) $f\_\theta$, and $\Psi: f\_\theta \mapsto \hat{I}$ as the mapping from the INR to the reconstructed image $\hat{I}$. Considering the adaptive attack, Eq. (2) should be written as
>
> $$
> \max\_{\Delta I\_{i}}\mathcal{L}\_{CE}(g\_{\phi}(\Psi\circ \Phi(I\_{i}+\Delta I\_{i})),y\_{i})\textrm{.}
> $$
> However, the mapping from the image $I$ to the intermediate representation (INR) $f\_\theta$ is obtained through optimizing Eq. (1). Therefore, the process $\Psi\circ \Phi$ is not amenable to gradient propagation. However, we can observe a property that $\Psi\circ \Phi(I)\approx I$. Based on the Backward Pass Differentiable Approximation (BPDA) method described in reference [1], we can estimate the gradient as $\nabla\_{I}\Psi\circ \Phi(I) \approx \nabla\_{I} I = 1$. Consequently, the above equation can be written as
>
> $$
> \max\_{\Delta I\_{i}}\mathcal{L}\_{CE}(g\_{\phi}(I\_{i}+\Delta I\_{i}),y\_{i})\textrm{.}
> $$
> This aligns with our formulation of Attack during creation in our paper. In fact, based on experimental results, this attack method is capable of successfully attacking methods utilizing conventional INR encoding. However, it can only be applied to the images undergoing encoding. Empirical evidence demonstrates that by considering additional adversarial loss during INR creation, we can effectively eliminate the perturbations added to the image.
>
> [1] Athalye et al. Obfuscated gradients give a false sense of security: Circumventing defenses to adversarial examples. ICML 2018
>
> > There are two extra computations introduced by the proposed method: first is the optimization process of $f\_{\theta}$, which is required to be optimized for each input image (i.e., cannot be amortized); second is the defense-in-creation process of Eq (4), which requires adversarial training by perturbing model parameters. The authors should report the empirical cost (e.g., computational time) for these two operations.
>
> The time for the first process is 7.12 seconds. The time for the second process is 175.50 seconds when no optimization for computational costs is considered. Besides, our setup does not need any time for training an adversarially robust downstream network model like previous methods. Since conventional adversarial
> training methods like PGD-AT take 3-30 times longer than standard training [1], the time consumption of our method is also acceptable.
>
> [1] Bai, Tao, et al. "Recent Advances in Adversarial Training for Adversarial Robustness." IJCAI 2021
>
> > The considered attacking methods such as PGD/DPGD, CW and FGSM are not strong. The authors should evaluate their methods under strong attacks like AutoAttack[3] and compare with the state-of-the-art models listed on RobustBench[4].
>
> References:
>
> [1] Athalye et al. Obfuscated gradients give a false sense of security: Circumventing defenses to adversarial examples. ICML 2018
>
> [2] Carlini et al. On evaluating adversarial robustness. arXiv 2019
>
> [3] Croce and Hein. Reliable evaluation of adversarial robustness with an ensemble of diverse parameter-free attacks. ICML 2020
>
> [4] https://robustbench.github.io/
>
> **Response:** The AutoAttack has not been specifically studied for INR. We conducted experiments by adapting AutoAttack for images during INR creation, and the results show that our defense-in-creation can reach high accuracy. For attack during transmission, due to the data being represented as INR, it is not feasible to directly apply AutoAttack to the INR parameters.

---

> > ### Author Response · Authors · 2023-11-21
> > **Response to reviewer Gy4W - Part 2**
> >
> > > From my experience, the proposed method (with 92.51% accuracy against PGD) probably has gradient obfuscation [1,2]. The authors should evaluate their method under adaptive attacks that involving the defense mechanism (i.e., INR mechanism), as well as strong off-the-shelf attacks such as AutoAttack.
> >
> > **Response:** As outlined in our paper, our method focuses on enhancing data-level adversarial robustness, and its motivation differs from traditional adversarial training approaches. In our envisioned scenario, the creators of INR, during the defense-in-creation process, are already aware of how the data will be used in a downstream network (such as certain fixed large-scale model architectures). In this case, their objective is to perform adversarial training on each INR, taking into account the knowledge of the downstream task network. This approach enables us to achieve an accuracy of 92.51%. During this process, gradient information is not obfuscated.
> >
> > Regarding adaptive attack during creation, since our defense-in-creation can adaptively eliminate adversarial perturbations present in the original image, we only consider adaptive attack during transmission. We employ PGD to apply perturbations to the reconstructed image and then fine-tune the resulting image with adversarial perturbations to obtain the final INR sample. Under this attack scenario, the accuracy on the CIFAR-10 dataset decreases to 84.47\% for 200-iteration fine-tuning and 2.00\% for 500-iteration fine-tuning.

---

### Official Review · Reviewer_DRHK · 2023-11-01

**Soundness:** 2 fair
**Presentation:** 3 good
**Contribution:** 2 fair
**Rating:** 3
**Confidence:** 5

**Summary:**

This paper investigates the problem of adversarial attacks in a new threat model designed around Implicit Neural Representations (INRs). INRs are a new family of data representations where each data point is represented via neural networks. To this end, often an MLP is used to overfit a data point. Then, the MLP weights can be used instead of each data point. Examples of INRs used in real applications include 2D images and 5D radiance fields (NeRFs).

In this context, this paper argues that INRs can be used to transmit data points between users, and as such, are prone to adversarial manipulation. In particular, an adversary might add a perturbation to the image data _before_ being encoded as INR, or they may opt to manipulate the INR weights _during transmission_. The former is called "attack during creation" while the latter is named "attack during transmission."

The paper presents a formulation for generating each of these attacks. Specifically, a projected gradient descent (PGD) attack is used to generate adversarial perturbations for attack during creation. Then, this adversarial example is encoded via an INR. Also, for attack during transmission a double projection gradient descent (DPGD) is used to ensure that while INR manipulation fools the downstream classifier, it generates attacks that are imperceptible in the image domain. Empirical evaluations indicate that both of these attacks are effective against CIFAR-10, CIFAR-100, and SVHN datasets.

Finally, as a potential defense against these attacks, the paper proposes to make the INR representation of the data robust to manipulation, resulting in an algorithm dubbed "defense in creation." To this end, the paper proposes a new objective function for creating the INR representation of the data by adding a regularization term. This term aims to add perturbations to the INR during creation while ensuring that these changes have minimal effect on the downstream classification task. Empirical results indicate that this strategy is helpful in making INRs robust to adversarial attacks.

**Strengths:**

- The paper investigates a new problem in adversarial machine learning. As far as this reviewer knows, there are no prior works that investigates robustness of INRs. Thus, the setting seems interesting, although its practicality needs to be discussed (see the Weaknesses).

- Setting the problem's definition aside, the paper explores the problem from different perspectives. In particular, exploring both the possible attacks and proposing a new defense strategy against them is rather appealing.

- Finally, the paper is very well-written and provides enough detail (such as figures and pseudo-codes) to understand each aspect of it.

**Weaknesses:**

- The major issue of this paper in this reviewer's view is its motivation and threat model. Specifically:

1. The paper starts its discussion by arguing that adversarial training is hard to deploy. The paper reads: "_Despite its effectiveness, adversarial training requires that users possess a detailed understanding of training settings._" In my view, this argument can even be used for using vanilla neural networks. One can make the same argument that even for using/training regular neural networks users need a detailed understanding of model architecture, optimization process, etc. There is no major differences that separates adversarial training from vanilla training, and I feel that motivating the core problem around such arguments is weak.

2. More importantly, the threat model is not intuitive. In particular, the paper assumes that one uses the INRs to encode the data, then send them to a model, which again decodes the INR to query a classifier. Why this process is efficient? Why the user doesn't send their data directly? What is special about this threat model? I believe that the proposes threat model is not making intuitive sense, and it requires a better design. For instance, using NeRF to encode a scene and then transmitting it would have made a much better threat model as NeRFs are encoding multiple scenes in one representation. However, using the current threat model for 2D images seems less intuitive and might not make any practical sense.

- The empirical evaluations are also limited. The paper only uses small scale datasets such as CIFAR and SVHN. It would be crucial to see how the proposed attacks and defense work in more large scale datasets such as ImageNet. Given that there are many pre-trained models available for the ImageNet dataset online, I believe that such evaluations would be feasible.

**Questions:**

- Please clarify what do you mean by arguments like "_our setup is rooted in the reality that many model users often fail to understand the settings associated with adversarial training._"

- Given that $\ell_p$ norms are used to enforce visual image similarity in the image domain, its use for other domains such as INR weights seems less intuitive. What are other alternatives for the similarity in the weight space that could have been used? In other words, are there any other alternatives to the $\ell_p$ norm used in Eq. (3)?

- What do you mean by $||\hat{I}\_{t-1}+\nabla-\hat{I}\_{\mathrm{org}}||_p \leq \zeta$? Is this a typo?

- It would be nice to add a few more sentences to the last paragraph of Section 3.2 explaining why projecting the gradient of the image space would help with having a better image quality. In other words, what motivated this step?

- What is the training time difference between finding an INR using Eq. (1) versus the defense in creation in Eq. (4)?

- Did the paper also test the transferability of the defense in creation? In other words, what happens if we find the robust INRs for a classifier $g\_{\boldsymbol{\phi}}^{(1)}$ while trying to defend another model $g\_{\boldsymbol{\phi}}^{(2)}$ during inference?

- Run experiments on large scale datasets such as ImageNet-1k or ImageNet-100.

- How do you specify the upper-bound for adversarial attack against INR weights? In other words, what makes a good $\delta$ for Eq. (3)? Because using $\ell_p$ norm in the INR weight space is not intuitive.

- In my view, using attack success rate (ASR) would be a better measure when trying to evaluate attacks. Using accuracy as the current version makes it difficult to interpret the results.

- How is it possible that the natural accuracies in Table 2 are 100%?

- Use a larger font size for the tables.


### Post-rebuttal Comments:
I want to sincerely thank the authors for their response.

As mentioned in my review, the proposed threat model makes no practical sense. The paper assumes that INRs instead of the query images are sent over the communication channels. Now, there are two major flaws with this threat model:

First, the data encoder has access to the target model (see the involvement of
 in Eqs. (3) and (4)). If the user had access to the target model, why don't they just use that model to run their inference? It doesn't make sense that the user creates a robust INR with the full knowledge of the target model, sends the data through a medium which is susceptible to adversarial attacks, and then runs inference on the same target model! As can be seen from the transferability results of this method, it is not transferable between architectures at all.

Second, having access to the true target label can even be considered as a serious flaw in the threat model. Why when we have the true label, we need to run any inference at all? Reading through the authors' response on a comment on this matter, I am still unsatisfied with the paper's approach.

Considering these two points and the authors' response, I decided to lower my initial score and recommend rejecting this paper. I hope that the authors can address these points in the future versions of their paper.

---

> ### Author Response · Authors · 2023-11-21
> **Response to reviewer DRHK - Part 1**
>
> > The paper starts its discussion by arguing that adversarial training is hard to deploy. The paper reads: "*Despite its effectiveness, adversarial training requires that users possess a detailed understanding of training settings. *" In my view, this argument can even be used for using vanilla neural networks. One can make the same argument that even for using/training regular neural networks users need a detailed understanding of model architecture, optimization process, etc. There are no major differences that separates adversarial training from vanilla training, and I feel that motivating the core problem around such arguments is weak.
>
> **Response:** We want to emphasize that adversarial training must consider potential attacks during model training or finetune the specific model to achieve adversarial robustness. However, when large-scale foundational model has become more popular today. Retraining specific models may become more difficult or even impossible [1]. In this situation, those model creators may define a specific data format suitable for their large-scale foundational models. Then, rather than directly retraining the model for adversarial robustness, our data robustness provides a more versatile solution. Besides, our experiments demonstrates that this unique robust data is able to achieve high robustness for specific models.
>
> [1] Hoffmann, Jordan, et al. "An empirical analysis of compute-optimal large language model training." Advances in Neural Information Processing Systems 35 (2022): 30016-30030.
>
> > Why this process is efficient? Why the user doesn't send their data directly?
>
> **Response:** This process may undermine the efficiency at the data side. However, it eliminates the need for time-consuming retraining or fine-tuning of the specific model. By incorporating the classifiers during the creation of INR, our method has achieved higher robustness for specific models in the current setting.
>
> If we directly send images, such incorporation can significantly undermine the image quality as shown in the "Pixel manipulation" experiment.
>
> > What is special about this threat model?
>
> **Response:** INR can encode the data into network parameters. By representing the data into INR, we are able to easily incorporate information of specific classifier into INR. Then, such unique data representation is able to achieve high robustness for specific models.
>
> Since we utilize INR for data representation, we need to consider perturbations to the INR parameters in the threat model.
>
> > Using NeRF to encode a scene and then transmitting it would have made a much better threat model as NeRFs are encoding multiple scenes in one representation. However, using the current threat model for 2D images seems less intuitive and might not make any practical sense.
>
> **Response:** NeRF is designed to encode a single, specific scene, utilizing multiple images from that scene for training. It is not capable of encoding multiple scenes within a single model. NeRF is also inapproproate for the tasks in our scenario requiring the representation of individual images. Besides, the use of INR for 2D image storage and transmission has been extensively studied. For example, [1] proposed that INR can achieve super-resolution image storage. [2] [3] considered using INR for image compression and transmission. [4] considered representing images as INR for neural network training. [5] implemented INR representation on the ImageNet dataset. Therefore, adopting INR as a cornerstone is practical and feasible.
>
> [1] Nguyen, Quan H., and William J. Beksi. "Single image super-resolution via a dual interactive implicit neural network." Proceedings of the IEEE/CVF Winter Conference on Applications of Computer Vision. 2023.
>
> [2] Strümpler, Yannick, et al. "Implicit neural representations for image compression." European Conference on Computer Vision. 2022.
>
> [3] Dupont, Emilien, et al. "COIN: COmpression with Implicit Neural representations." Neural Compression: From Information Theory to Applications--Workshop@ ICLR 2021. 2021.
>
> [4] Rusu, Andrei A., et al. "Hindering Adversarial Attacks with Implicit Neural Representations." International Conference on Machine Learning. PMLR, 2022.
>
> [5] Singh, Rajhans, Ankita Shukla, and Pavan Turaga. "Polynomial Implicit Neural Representations For Large Diverse Datasets." Proceedings of the IEEE/CVF Conference on Computer Vision and Pattern Recognition. 2023.

---

> > ### Author Response · Authors · 2023-11-21
> > **Response to reviewer DRHK - Part 2**
> >
> > > The empirical evaluations are also limited. The paper only uses small scale datasets such as CIFAR and SVHN. It would be crucial to see how the proposed attacks and defense work in more large-scale datasets such as ImageNet. Given that there are many pre-trained models available for the ImageNet dataset online, I believe that such evaluations would be feasible.
> >
> > **Response:** We further conduct experiment on ImageNet. We used pre-trained weights from the publicly available torchvision.models package, specifically the 'ResNet50_Weights.IMAGENET1K_V2'. The result on ImageNet is shown as below. Our method keeps robust performance.
> >
> > |                     | DPGD    | PGD for INR | CW for INR |
> > |---------------------|---------|-------------|------------|
> > |   Normal training   | 12.25\% | 12.88\%     | 1.88\%     |
> > | Defense-in-creation | 98.00\% | 99.87\%     | 83.75\%    |
> >
> > > Please clarify what do you mean by arguments like "*our setup is rooted in the reality that many model users often fail to understand the settings associated with adversarial training. *"
> >
> > **Response:** In our paper, our goal is to transfer data into robust form for publicly available models (they could be large-scale pretrained models in the future). Since it is difficult to retrain all these models adversarially, these models may lack the ability to defend against adversarial attacks. Therefore, traditional data form can be easily perturbated to fool these models. Consequently, we encode the data into INR parameters incorporating specific classifier information for robust data representation. We will change this sentence to:
> >
> > "*Our setup is rooted in the reality that many publicly available models lack the ability to defend against adversarial attacks, and training them adversarially is difficult.*"
> >
> > > Given that $\ell\_p$ norms are used to enforce visual image similarity in the image domain, its use for other domains such as INR weights seems less intuitive. What are other alternatives for the similarity in the weight space that could have been used? In other words, are there any other alternatives to the $\ell\_p$ norm used in Eq. (3)?
> >
> > **Response:** Using $\ell\_p$ norm in parameter space between networks is a common setting [1]. In our experiments, to maintain the efficiency and simplicity of the whole framework, we directly use the $\ell\_p$ norm, which has already shown promising performance. We agree that there is no consensus on the definition of network similarity [2]. In the future, we will explore other methods to measure weights similarity for better performance.
> >
> > [1] Bansal, Arpit, et al. "Certified neural network watermarks with randomized smoothing." International Conference on Machine Learning. PMLR, 2022.
> >
> > [2] Kornblith, Simon, et al. "Similarity of neural network representations revisited." International conference on machine learning. PMLR, 2019.
> >
> > > What do you mean by $||\hat{I}\_{t-1}+\nabla-\hat{I}\_{\mathrm{org}}||\_p \leq \zeta$? Is this a typo?
> >
> > **Response:** For clarity, we rewrite line 5 to line 9 in Algorithm 2 as
> >
> > 5: Reconstruct image $\hat{I}\_{t-1} = \Psi(f\_{\theta\_{t-1}})$
> >
> > 6: $Grad\_{I} = \frac{\partial \mathcal{L}\_{CE}(g_{\phi}\left(\hat{I}\_{t-1}\right), y)}{\partial \hat{I}\_{t-1}}$
> >
> > 7: Update $Grad\_{I} = \operatorname{Proj}\_{\left\|\hat{I}\_{t-1} +Grad\_{I} - \hat{I}\_{org}\right\|\_p \leq \zeta}(Grad\_{I})$
> >
> > 8: $Grad\_{\theta} = \frac{\partial \hat{I}\_{t-1}}{\partial \theta\_{t-1}} \cdot Grad\_{I}$
> >
> > 9: $\theta_t=\theta_{t-1}+\alpha \cdot \operatorname{sign}(Grad_{\theta})$
> >
> > The $Grad\_{I}$ refers to the gradient on the image pixels, while the $Grad\_{\theta}$ refers to the gradient on INR parameters. This formula appears in Algorithm 2 as the first projection. We will explain the motivation and significance of doing so in the next question.

---

> > > ### Author Response · Authors · 2023-11-21
> > > **Response to reviewer DRHK - Part 3**
> > >
> > > > It would be nice to add a few more sentences to the last paragraph of Section 3.2 explaining why projecting the gradient of the image space would help with having a better image quality. In other words, what motivated this step?
> > >
> > > **Response:** In PGD attacks against images, the perturbation is projected to a specified pixel range in each iteration. However, attacks on INR do not directly manipulate the image. If only the fluctuation range of INR parameters is considered, the image decoded from the perturbed INR could be severely damaged. Therefore, we consider adding constraints directly on the image level for the gradients. The detailed analysis is as follows. The loss function is defined as $ J(\theta) = \mathcal{L}\_{CE}(g\_{\phi}(\hat{I}), y) $. Here, $ \hat{I} = \Psi(f\_\theta) $ represents the image reconstructed by INR $ f\_\theta $. The original input's INR weight is $ \theta\_{org} $, and the image it reconstructs is $ \hat{I}\_{org} $. In the $ t $-th iteration, the image reconstructed by the INR with weights $ \theta\_{t-1} $ before the attack is $ \hat{I}\_{t-1} $, and the loss is $ J(\theta\_{t-1}) $. The gradient of the loss at $ \theta\_{t-1} $ is calculated as $ \nabla\_{\theta\_{t-1}} = \frac{\partial J}{\partial \theta\_{t-1}} $. Updating against the gradient direction, we get $ \theta\_{t} = \theta\_{t-1} + \alpha \frac{\partial J}{\partial \theta\_{t-1}} $. At this point, the new loss $ J' $ can be estimated as:
> > >
> > >
> > > $$
> > > J(\theta\_{t}) \approx J(\theta\_{t-1}) + (\theta\_{t}-\theta\_{t-1})\frac{\partial J}{\partial \theta\_{t-1}}
> > > $$
> > > $$
> > >  = J(\theta\_{t-1}) + \alpha (\frac{\partial J}{\partial \theta\_{t-1}})^2.
> > > $$
> > > Since the second term in the equation above is always positive, it allows for updates in a direction that increases the loss gradually. Now, we consider the impact of weight updates on the reconstructed image. The reconstructed image can be approximated as
> > > $$
> > > \hat{I}\_{t} \approx \hat{I}\_{t-1} + (\theta\_{t}-\theta\_{t-1})\frac{\partial \hat{I}}{\partial \theta\_{t-1}}
> > > $$
> > > $$
> > > =\hat{I}\_{t-1} + \alpha \frac{\partial J}{\partial \theta\_{t-1}}\frac{\partial \hat{I}}{\partial \theta\_{t-1}}
> > > $$
> > > $$
> > > =\hat{I}\_{t-1} + \alpha \frac{\partial J}{\partial \hat{I}} \frac{\partial \hat{I}}{\partial \theta\_{t-1}} \frac{\partial \hat{I}}{\partial \theta\_{t-1}}
> > > $$
> > > $$
> > > = \hat{I}\_{t-1} + \alpha (\frac{\partial \hat{I}}{\partial \theta\_{t-1}})^2 \frac{\partial J}{\partial \hat{I}}.
> > > $$
> > > To prevent significant differences between the reconstructed image and the original image, we project $\hat{I}{t}$ onto the $\ell\_p$ ball around the original reconstructed image $\hat{I}{org}$. Therefore, the following constraint is added:
> > > $$
> > > ||\hat{I}\_{t}-\hat{I}\_{\mathrm{org}}||\_p \leq \zeta.
> > > $$
> > > That is
> > > $$
> > > ||\hat{I}\_{t-1} + \alpha (\frac{\partial \hat{I}}{\partial \theta\_{t-1}})^2 \frac{\partial J}{\partial \hat{I}}-\hat{I}\_{\mathrm{org}}||\_p \leq \zeta.
> > > $$
> > > Since $\alpha (\frac{\partial \hat{I}}{\partial \theta\_{t-1}})^2$ is always positive, for ease of computation, we simplify it as $\alpha (\frac{\partial \hat{I}}{\partial \theta\_{t-1}})^2=1$, which leads to line 7 in Algorithm 2.
> > >
> > > We will add an explanation to the last paragraph of Section 3.2 in the revised paper.
> > >
> > > > What is the training time difference between finding an INR using Eq. (1) versus the defense in creation in Eq. (4)?
> > >
> > > **Response:** With our experimental setting, the training time using Eq. (1) is 7.12 seconds, while the training time using Eq. (4) is 175.50 seconds. Since conventional adversarial training methods like PGD-AT take 3-30 times longer than standard training [1], the time consumption of our method is also acceptable.
> > >
> > > [1] Bai, Tao, et al. "Recent Advances in Adversarial Training for Adversarial Robustness." IJCAI 2021
> > >
> > >
> > > > Did the paper also test the transferability of the defense in creation? In other words, what happens if we find the robust INRs for a classifier $g\_{\boldsymbol{\phi}}^{(1)}$ while trying to defend another model $g\_{\boldsymbol{\phi}}^{(2)}$ during inference?
> > >
> > > **Response:** Our method is model-specific, which means that we aim at producing robust data with high robustness for specific models. Thus, the transferability is beyond our setup. For example, we train our robust INRs for a classifier using PreActResNet-18 architecture, and test on WideResNet34-10 architecture. The accuracy under CW attack is 22.67\%, higher than normal training INR (12.80\%), but much lower than accuracy using the same PreActResNet-18 architecture. However, the users can train with different models simutenously when generating robust INRs. Then, the robust data can have robust performance on those specific models incorporated during INR generation.
> > >
> > > > Run experiments on large scale datasets such as ImageNet-1k or ImageNet-100.
> > >
> > > **Response:** We have addressed this in the above experiments.

---

> > > > ### Author Response · Authors · 2023-11-21
> > > > **Response to reviewer DRHK - Part 4**
> > > >
> > > > > How do you specify the upper-bound for adversarial attack against INR weights? In other words, what makes a good $\delta$ for Eq. (3)? Because using $\ell\_p$ norm in the INR weight space is not intuitive.
> > > >
> > > > **Response:** As said in our paper, we evaluate the attack using two criteria: one is invisibility of perturbations in the reconstructed image, and the other is the attacking efficacy of the perturbations. We believe that a good $\delta$ should result in an INR-reconstructed image that is as consistent as possible with the original image while being able to fool the downstream applications. In our experiments, we tested different values of $\delta$, and the results are shown in Figure 4 (right). The horizontal axis represents the PSNR between the reconstructed image and the original image, while the vertical axis shows the accuracy after the attack. A good $\delta$ should strike a balance between invisibility and attacking efficacy.
> > > >
> > > > > In my view, using attack success rate (ASR) would be a better measure when trying to evaluate attacks. Using accuracy as the current version makes it difficult to interpret the results.
> > > >
> > > > **Response:**  Thank you for your suggestion. We will use attack success rate (ASR) for evaluating attacks. We will modify the first two rows in Table 1 as
> > > >
> > > > | Method | FGSM on images | Creation after FGSM | PGD on images | Creation after PGD | CW on images | Creation after CW |
> > > > |--------|----------------|---------------------|---------------|--------------------|--------------|-------------------|
> > > > | ASR    | 60.45\%        | 59.26\%             | 100.00\%      | 99.83\%            | 99.93\%      | 99.84\%           |
> > > >
> > > > We will also make revisions to Figure 4 in our paper.
> > > >
> > > > > How is it possible that the natural accuracies in Table 2 are 100%?
> > > >
> > > > **Response:** In our setup, by incorporating the specifical classifier during the creation of INR, we create this envisioned data format that is able to achieve envisioned high data robustness on specific models Thus, they can reach $100\%$. You can see more details from the demo in our supplementary. Since our setting differs from the traditional approach, to avoid ambiguity, we will changed the "Natural" to "Accuracies without attacks."
> > > >
> > > > > Use a larger font size for the tables.
> > > >
> > > > **Response**: Thanks for your suggestion. We will use a larger font size in our revised paper.

---

### Author Response · Authors · 2023-11-21
**Thanks for your feedback**

Dear Reviewers,

Thank you sincerely for taking the time to provide feedback on our work. We are grateful for your valuable comments and suggestions. Should you have further questions or insights regarding our rebuttal or our work, we kindly invite you to share them with us. Your guidance is truly invaluable, and we sincerely appreciate your support.

Best Regards,

Authors

---

### Author Response · Authors · 2023-11-22
**Response to the public comment**

We received an e-mail notifying us of a new comment on our submission. Although the commenter deleted his/her comment, we still want to explain more on that.

**Comment:** For each image in test set, we don't have the label information, how can we supervise the training of the robust INR from an INR with adversarial perturbations?

**Response:** First, our goal is to build robust data format for specific models. Our method can incorporate information of specific model into INR to enhance higher robustness. In this situation, model developers create a number of robust data via our method and model users can only use those safe data for their purposes. As model developers are willing to provide model information and label to create data with high robustness for specific models, the concerns for the lack of label actually does not exist in our setting, since we do not differentiate the test data and training data. Second, the adversarial attack to INR is performed after the creation of INR, so we do not train from an INR with adversarial perturbations.

**Comment:** In my opinion, the adversary provides an adversarial INR, this paper modify it to make the reconstructed data clean with the ground truth label. It's like that we use ground truth label to adjust an adversarial sample to make it innocuous while INR is dispensable.

**Response:** Our method is not simply modifying the INR to make it clean. INR can encode the data into network parameters, and we can incorporate information of specific classifier into these parameters. This process can make the INR defend against both attacks during creation and attacks during transmission. INR is not dispensable. We also try to manipulate the image pixels directly, the results in Table 2 in our submission show that such method undermines image quality and cannot defend against attacks during transmission effectively. Due to its network nature, INR is able to better incorporate information from a specific classifier.

---

> ### Public Comment · ~Binxiao_Huang1 · 2023-11-23
> **Question about lack of label**
>
> First of all, thanks for your work in the paper. I have a question about "Proposed defense method to produce INR against adversarial attacks".
>
> Since the downstream model is fixed, given any image in test dataset, you can generate an adversarial INR to make the downstream model fail (Figure 2). To make the INR robust, you train the INR using the **robustness loss** with the real label (line 5 in algorithm3), which is unavailable for image in test dataset.
>
> If $\lambda_2$ in equation (4) is set to zero (no information about real label), the accuracy is quite low (Table 4 in appendix). Besides, the accuracy is almost **100%** for FGSM/PGD in INR, even higher than the normal input. Isn't that strange?

---

> > ### Author Response · Authors · 2023-11-23
> > **Response to the public comment**
> >
> > Thank you for your comments.
> >
> > Our setting is different from traditional adversarial defense. Our target is to produce INR against adversarial attacks for the specific model. The model developer uses the model and label to train the INR, and release these INRs to the public. Through our method, the released INRs can defend against further adversarial attacks.
> >
> > Our accuracy can be high as we enhance the data robustness for specific pre-trained models. In our scenario, model developers are willing to provide model information and labels to create data with high robustness for specific models. As more information is incorporated into the INR parameters during training, our results can be higher than the normal input without additional information.

---

> > > ### Author Response · Authors · 2023-11-23
> > > **Response to the public comment**
> > >
> > > For $\lambda_2$ you mentioned, this hyperparameter is used to strike a balance between the reconstruction quality and robustness. When $\lambda_2$ is set to $0$, it implies regular INR training where the robustness problem is not intentionally considered. As a result, the accuracy is low when the data is attacked during transmission. On the other hand, our method incorporates classifier information during INR creation by setting $\lambda_2 > 0$, which enables us to achieve high accuracy even during transmission.

---

> > > > ### Author Response · Authors · 2023-11-23
> > > > **Response to the public comment**
> > > >
> > > > Our robust data is model-specific and can only be published by the model developer. Therefore, labeling is not a problem here. By utilizing this type of data, we can ensure a high level of security for the model. This is why we can directly use the labels from the test set to generate robust data and study its robustness. In our scenario, the release of data is strictly controlled by the model developer.

---

> > > ### Public Comment · ~Binxiao_Huang1 · 2023-11-23
> > > **Follow-up question**
> > >
> > > Thank you for your response.
> > >
> > > The released INRs are robust because this paper modify it according to the true label. For any input, it can only guarantee its robustness if you know its true label.
> > > In most cases, the attacker provide an adversarial INR to make the downstream task fail, while the defender has no idea about the true label. For example, the attacker provides an adversarial INR representing a CAT to let the model classify, the defender can not get any more information, except for the INR, how can you make the INR robust to the adversarial attacks?

---

> ### Author Response · Authors · 2023-11-23
> **Response to the follow-up question**
>
> Thank you for your question.
>
> Our goal is to produce robust data for specific models. These data can be collected as a robust database. Our method can make the data in the database robust against adversarial attacks. In your case, if the attacker provides an adversarial INR beyond this database, it can be easily detected that this data does not belong to the database.

---

> > ### Public Comment · ~Binxiao_Huang1 · 2023-11-23
> > **Thanks for your response.**
> >
> > Thanks a lot for your patient explanation.
> >
> > The finite robust INRs are built based on the premise that the real labels are available. Any data (without a label) outside of the robust dataset is still vulnerable to the adversarial attack.
> >
> > Sorry to bother you so much time, good luck to you!

---

### Meta-Review · Area_Chair_UrWK · 2023-12-05

**Metareview:**

This paper sets out to improve adversarial data robustness by introducing Implicit Neural Representation (INR). The gist of the method is to apply an overparameterized neural network representation to the input. The argument is that the INRs can be transmitted across users and may be subject to adversarial attacks. The paper proposes two types of attacks: (1) attack during creation: where an adversary might add a perturbation to the image data before INR encoding; (2) attack during transmission: where an adversary may manipulate the INR weights during transmission after encoding. The paper adapts variants of projected gradient descent (PGD) to curate attacks on INRs and empirically verifies their effectiveness on CIFAR-10, CIFAR-100, and SVHN datasets. The paper also proposes a defense that inttrdicues a robustness regularizer during INR encoding, and empirically shows that this method is effective.

While there was initial excitement about the work, during the discussion all reviewers and the AC realized that INR uses target label information, which makes the proposed method impractical and questions the motivation for the work. As such, *all* reviewers and AC agreed during the discussion that the paper should be rejected due to this serious flaw. We hope that the authors can address this serious flaw in a future iteration and resubmit their paper addressing the reviewers' comments.

**Justification For Why Not Higher Score:**

The paper suffers from a serious flaw that in non-trivial to fix.

**Justification For Why Not Lower Score:**

The subject area of the paper is quite exciting, and the developed attack/defense methods are quite novel and interesting modulo the serious design flaw.

---

### Decision · Program_Chairs · 2024-01-16

Reject